# HARMONY WORLD MODELS:
# BOOSTING SAMPLE EFFICIENCY FOR MODEL-BASED REINFORCEMENT LEARNING

## ABSTRACT

Model-based reinforcement learning (MBRL) holds the promise of sample-efficient learning by utilizing a world model, which models how the environment works and typically encompasses components for two tasks: observation modeling and reward modeling. In this paper, through a dedicated empirical investigation, we gain a deeper understanding of the role each task plays in world models and uncover the overlooked potential of more efficient MBRL by mitigating the domination of either observation or reward modeling. Our key insight is that while prevalent approaches of explicit MBRL attempt to restore abundant details of the environment through observation models, it is difficult due to the environment's complexity and limited model capacity. On the other hand, reward models, while dominating implicit MBRL and adept at learning task-centric dynamics, are inadequate for sample-efficient learning without richer learning signals. Capitalizing on these insights and discoveries, we propose a simple yet effective method, Harmony World Models (HarmonyWM), that introduces a lightweight harmonizer to maintain a dynamic equilibrium between the two tasks in world model learning. Our experiments on three visual control domains show that the base MBRL method equipped with HarmonyWM gains $10\%-55\%$ absolute performance boosts.

## 1 INTRODUCTION

Learning efficiently to operate in environments with complex observations requires generalizing from past experiences. Model-based reinforcement learning (MBRL, Sutton (1990)) utilizing world models (Ha & Schmidhuber, 2018; LeCun, 2022) offers a promising approach. In MBRL, the agent learns behaviors by simulating trajectories based on predictions generated by the world model. Concurrently, the world model itself is designed to learn two key components of dynamics (defined in Sec. 2.1): how the environment transits and is observed (i.e. the *observation modeling* task) and how the task has been progressed (i.e. the *reward modeling* task) (Kaiser et al., 2020; Hafner et al., 2020; 2021). As imaginary rollouts reduce the need for real environment interactions, world models enable model-based RL agents to outperform their model-free counterparts in terms of sample efficiency.

While observation transitions and rewards in low-dimensional spaces can be classically learned by separate models, for environments with high-dimensional and partial observations, it is favorable for world models to learn both tasks from a shared representation, a form of *multi-task learning*, aiming to improve learning efficiency and generalization performance (Jaderberg et al., 2017; Laskin et al., 2020; Yarats et al., 2021). However, to best exploit the benefits of multi-task learning, it demands careful designs to weigh the contribution of each task without allowing either one to dominate (Misra et al., 2016; Kendall et al., 2018), which naturally leads to the following question:

*Do existing MBRL methods properly exploit the multi-task benefits in world model learning?*

In this work, we take a novel and unified multi-task perspective to revisit world model learning in MBRL literature (Moerland et al., 2023): Prevalent approaches, named as *explicit* MBRL (Kaiser et al., 2020; Hafner et al., 2021; Seo et al., 2022b), aim to learn an exact duplicate of the environment by predicting each individual element (e.g., observations, rewards, and terminal signals), which gives the agent access to accurately learned transitions. However, learning to predict future observations can be difficult and inefficient since it encourages the world model to model everything in the environment,

Figure 1: A multi-task perspective of world models. (*Left*) World models typically consist of components for two tasks: **observation modeling** and **reward modeling**. (*Right*) A spectrum of world model learning in MBRL. Explicit MBRL learns models dominated by observation modeling, while implicit MBRL relies solely on reward modeling. Our Harmony World Models provide an explicit-implicit solution that maintains a dynamic equilibrium between them to unleash the multi-task benefits of world model learning, thus boosting the sample efficiency of MBRL.

including task-irrelevant nuances (Okada & Taniguchi, 2021; Deng et al., 2022). Consequently, world model learning in explicit MBRL is typically dominated by observation modeling to capture complex observations and their associated dynamics but still suffers from model inaccuracies and compounding errors. Meanwhile, reward modeling, which only predicts an additional scalar, is commonly overlooked. Another line of work, known as *implicit* MBRL, from a task-centric view, learns world models solely from reward modeling (Oh et al., 2017; Schrittwieser et al., 2020; Hansen et al., 2022) to realize the value equivalence principle, i.e., the predicted rewards along a trajectory of the world model matches that of the real environment (Grimm et al., 2020). This approach builds world models directly useful for MBRL to identify the optimal policy or value, and tends to perform better in tasks where the complete dynamics related to observations are too complicated to be perfectly modeled. Nevertheless, as the reward signals in RL are known to be sparser than signals in supervised learning, potentially leading to representation learning challenges, it is more practical to incorporate auxiliary tasks that provide richer learning signals beyond rewards (Jaderberg et al., 2017).

To support above insights, our dedicated empirical investigation reveals surprising deficiencies in sample efficiency for the default practice of a state-of-the-art model-based method (Dreamer, Hafner et al. (2020; 2021; 2023)), as *increasing the coefficient of reward loss in world model learning leads to dramatically boosted sample efficiency* (see Sec. 2.3). We identify the root cause as the domination of observation models in explicit world model learning: due to an overload of redundant observation signals, the model may establish spurious correlations in observations without realizing incorrect reward predictions, which ultimately hinders the learning process of the agent. On the other hand, implicit MBRL, which learns world models solely exploiting reward modeling, is also proven to be inefficient. In summary, while widely adopted in existing MBRL literature, domination of either task cannot properly exploit the multi-task benefits in world model learning.

As shown in Fig. 1, we propose to address the problem with Harmony World Models (HarmonyWM), a simple *explicit-implicit* approach for world model learning that exploits the advantages of both sides. Concretely, HarmonyWM introduces a lightweight harmonizer to maintain a dynamic equilibrium between reward and observation modeling during world model learning. We evaluate our approach on various challenging visual control domains, including Meta-world (Yu et al., 2020b), RLBench (James et al., 2020), and DMC Remastered (Grigsby & Qi, 2020), and demonstrate that HarmonyWM consistently promotes sample efficiency and also owns generality to base MBRL approach.

**Our contributions.** In summary, we make the following key contributions:

- To the best of our knowledge, our work, for the first time, systematically identifies the multi-task essence of world models and analyzes the deficiencies caused by the domination of a particular task, which is unexpectedly overlooked by most previous works.
- We propose the Harmony World Model (HarmonyWM), a simple explicit-implicit world model learning approach to mitigate the domination of either observation or reward modeling, without the need for exhaustive hyperparameter tuning.
- Our experiments show that HarmonyWM can significantly boost the sample efficiency of MBRL on various challenging domains. Measured in terms of policy success rates or returns, Dreamer equipped with HarmonyWM achieves $10\% - 55\%$ higher **absolute** performance

across domains (up to 74% more success on the Meta-world Push task). We also demonstrate the generality of HarmonyWM for other base MBRL methods (Deng et al., 2022).

## 2 A MULTI-TASK PERSPECTIVE OF WORLD MODELS

In this paper, we focus on vision-based RL tasks, which can be formulated as partially observable Markov decision processes (POMDP). A POMDP is defined as a tuple $(\mathcal{O}, \mathcal{A}, p, r, \gamma)$, where actions $a_t \sim \pi(a_t \mid o_{\leq t}, a_{<t})$ generated by the agent receive high-dimensional observations $o_t \sim p(o_t \mid o_{<t}, a_{<t})$ and scalar rewards $r_t = r(o_{\leq t}, a_{<t})$ generated by the unknown transition dynamics $p$ and reward function $r$ of the environment. The goal of MBRL is to learn an agent that maximizes the $\gamma$-discounted cumulative rewards $\mathbb{E}_{p,\pi}\left[\sum_{t=1}^{T} \gamma^{t-1} r_t\right]$, leveraging a learned world model which approximates the underlying environment $(p, r)$.

### 2.1 TWO TASKS IN WORLD MODELS

Two key tasks can be identified in world models, namely observation and reward modeling.

**Definition 2.1.** *The **observation modeling** task in world models is to predict consequent observations $p(o_{t+1:T} \mid o_{1:t}, a_{1:T})$ of a trajectory, given future actions. Similarly, the **reward modeling** task in world models is to predict future rewards $p(r_{t+1:T} \mid o_{1:t}, a_{1:T})$.*

As mentioned before, these two tasks provide a unified view of MBRL: while explicit MBRL learns world models for both observations and rewards to mirror the complete dynamics of the environment, implicit MBRL only learns from reward modeling to capture task-centric dynamics.

### 2.2 OVERVIEW OF WORLD MODEL LEARNING

We conduct detailed analysis and build our method primarily upon DreamerV2 (Hafner et al., 2021), but we also demonstrate the generality of our method to various base MBRL algorithms, including DreamerV3 (Hafner et al., 2023) and DreamerPro (Deng et al., 2022) (see Sec. 4.4).

The world model in Dreamer (left in Fig. 1) consists of the following four components:

$$
\begin{array}{llll}
\text{Representation model:} & z_t \sim q_\theta(z_t \mid z_{t-1}, a_{t-1}, o_t) & \text{Observation model:} & \hat{o}_t \sim p_\theta(\hat{o}_t \mid z_t) \\
\text{Transition model:} & \hat{z}_t \sim p_\theta(\hat{z}_t \mid z_{t-1}, a_{t-1}) & \text{Reward model:} & \hat{r}_t \sim p_\theta(\hat{r}_t \mid z_t)
\end{array} \tag{1}
$$

The latent representation $z_t$ is generated by the representation model using the previous latent state $z_{t-1}$, the current action $a_{t-1}$, and the current visual observation $o_t$. The latent prediction $\hat{z}_t$, meanwhile, is generated by the transition model using only the previous state and current action. All model parameters $\theta$ are trained to jointly learn the observations, rewards, and transitions of the environment by minimizing the following objectives:

$$
\begin{array}{lll}
\text{Observation loss:} & \mathcal{L}_o(\theta) = -\log p_\theta(o_t \mid z_t) \\
\text{Reward loss:} & \mathcal{L}_r(\theta) = -\log p_\theta(r_t \mid z_t) \\
\text{Dynamics loss:} & \mathcal{L}_d(\theta) = \mathrm{KL}\left[q_\theta(z_t \mid z_{t-1}, a_{t-1}, o_t) \,\|\, p_\theta(\hat{z}_t \mid z_{t-1}, a_{t-1})\right],
\end{array} \tag{2}
$$

where the dynamics loss simultaneously trains the latent predictions toward the representations, and regularizes the representations to be predictable. In practice, the observation model and reward model typically leverage Gaussian distributions, and both losses take the form of a simple $L_2$ loss between prediction $\hat{o}_t, \hat{r}_t$ and ground truth $o_t, r_t$ respectively, excluding irrelevant constants.

Taking our multi-task perspective, the observation model and reward model with their associated losses account for the aforementioned two tasks in the world model of Dreamer. However, they do not operate in isolation, and instead interact with and regularize each other upon the shared representation and transition model, in pursuit of either complete or task-centric latent dynamics, respectively.

The overall objective of world model learning can be formulated as follows:

$$
\mathcal{L}(\theta) = w_o \mathcal{L}_o(\theta) + w_r \mathcal{L}_r(\theta) + w_d \mathcal{L}_d(\theta). \tag{3}
$$

By default, $w_o$, $w_r$, and $w_d$ are typically set to approximately equal weights (namely, $w_o = w_r = w_d = 1.0$) (Hafner et al., 2020; 2021; Seo et al., 2022b; Wu et al., 2022), overlooking the potential domination of a particular task. In contrast, we conduct a careful empirical investigation to understand the role each task plays in world models and reveal the deficiency of the default weighting practice.

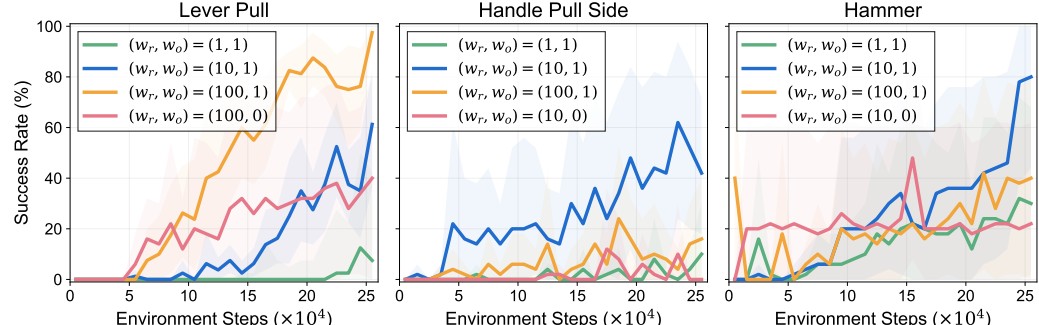

Figure 2: Effects of different loss coefficients in world model learning for the DreamerV2 agent on Meta-world tasks. Simply adjusting the coefficient of reward loss leads to dramatically boosted sample efficiency, indicating the potential multi-task benefits yet to be properly exploited.

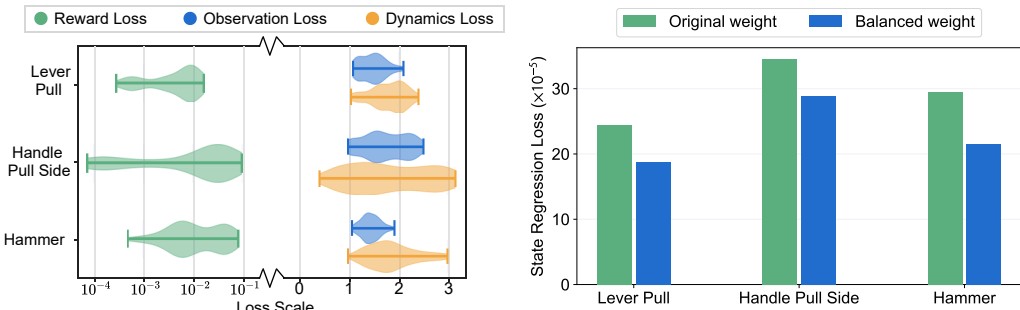

Figure 3: Scales of the three losses in world model learning. The reward loss is shown to be two orders of magnitude smaller than the others.

Figure 4: State regression loss measuring the ability of world models' representations to predict the ground truth environment states.

## 2.3 DIVE INTO WORLD MODEL LEARNING

We consider the tasks of *pulling a lever up*, *pulling a handle up sideways*, and *hammering a screw on the wall*, from the Meta-world domain (Yu et al., 2020b), as our testbed to investigate world model learning. The prominent improvements of the derived approach in our benchmark experiments (see Sec. 4) prove that our discoveries can be generalized to various domains and tasks.

First of all, we experiment with simply adjusting the coefficient of the reward loss in Eq. (3). Results in Fig. 2 reveal a surprising fact that by simply tuning the reward loss weights ($w_r \in \{1, 10, 100\}$), the agent can achieve considerable improvements in terms of sample efficiency.

> **Finding 1.** *Leveraging the reward loss by adjusting its coefficient in world model learning has a great impact on the sample efficiency of model-based agents.*

One obvious reason for this is that the reward loss only accounts for a tiny proportion of the learning signals, actually a single scalar $r_t$. As shown in Fig. 3, the scale of $\mathcal{L}_r$ is two orders of magnitude smaller than that of $\mathcal{L}_o$, which usually aggregates $H \times W \times C$ dimensions. As discussed before, reward modeling is crucial for extracting task-relevant representations and driving behavior learning of the agents. Dominated by the observation modeling task, the world model fails to learn a task-centric latent space and predict accurate rewards, which hinders the learning process of the agent.

We then explore further to demonstrate how the observation modeling task dominating world models can specifically hurt behavior learning. To isolate distracting factors, we consider an offline setting (Levine et al., 2020). Concretely, we use a fixed replay buffer on the task of Lever Pull and offline train DreamerV2 agents with different reward loss coefficients on it (see details in Appendix C.4). In Fig. 5, we showcase a trajectory where the default Dreamer agent ($w_r = 1$) fails to lift the lever. It is evident that it learns a spurious correlation (Geirhos et al., 2020) between the actions of the robot and that of the lever and predicts inaccurate transitions and rewards, which misleads the agents to unfavorable behaviors. Properly balancing the reward loss ($w_r = 100$) can emphasize task-relevant information,

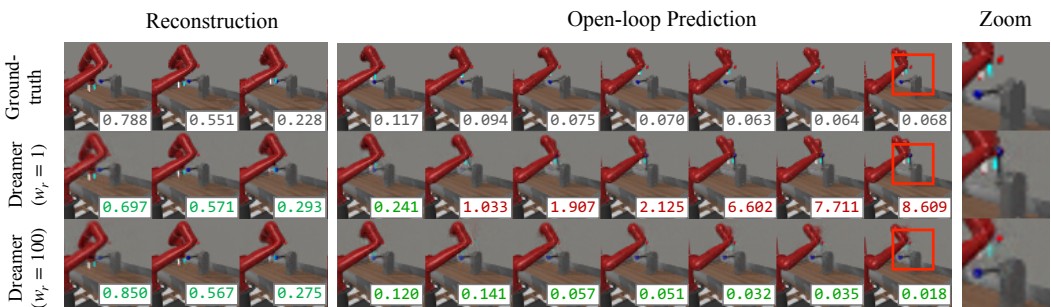

Figure 5: Analysis of world models learned with different reward loss coefficients. Rewards are labeled at the bottom right corner, with predictions marked as **correct** or **incorrect**. Dominating observation modeling in world models incurs spurious correlations between actions, observations, and rewards, which can be dissolved by properly emphasizing reward modeling.

such as whether the lever is actually lifted, to correct hallucinations by world models. Quantitative analysis in Fig. 4 measuring the ability of world models' representations to predict the ground truth states also suggests emphasizing reward modeling learns better task-centric representations.

> **Finding 2.** *Observation modeling as a dominating task can result in world models establishing spurious correlations without realizing incorrect reward predictions.*

Although we have shown above that exploiting reward modeling can bring benefits to world models and MBRL, it might be over-demanding for a world model to learn solely by rewards, as implicit MBRL. As we discussed before, learning world models depending fully on scarce reward signals has limited capability to learn meaningful representations, and thus can encounter optimization challenges and hinder sample-efficient learning (Yarats et al., 2021). Our experiment results in Fig. 2 show that the disuse of the observation loss in DreamerV2 produces inferior results with a high variance.

> **Finding 3.** *Learning signal of world models from rewards alone without observations is inadequate for sample-efficient learning.*

**Discussion.** We are not the first to adjust loss coefficients in world model learning, but we dedicatedly investigate this. Here we discuss the differences between our findings and previous literature. Our Finding 1 coincides with high reward loss weights manually tuned (typically 100 or 1000) in decoder-free model-based RL (Nguyen et al., 2021; Deng et al., 2022). Our analysis differs from theirs in two significant ways: 1) We focus on a decoder-based world model, where the observations are learned from explicit reconstructions. 2) We discovered that emphasizing reward modeling is also beneficial for visually simple tasks (e.g. Meta-world tasks), in addition to visually demanding tasks with noisy backgrounds. Our Finding 3 is similar to the reward-only ablation in Dreamer (Hafner et al., 2020), but we prove that even if given higher loss weights, learning a world model purely from rewards is less sample-efficient than properly exploiting both observation and reward modeling.

## 3 HARMONY WORLD MODELS

In light of the discoveries and insights, we propose a simple yet effective method as the first step towards exploiting the multi-task essence of world models. Instead of disharmonious domination, we aim to build a harmonious interaction between the two tasks in world models: while observation modeling facilitates representation learning and prevents information loss, reward modeling enhances task-centric representations to correctly inform behavior learning of the agents.

As shown in Fig. 6, we build our method on Dreamer (Hafner et al., 2020; 2021), and brand it as Harmony World Models (HarmonyWM), for it mitigates potential domination of a particular task in world models. Specifically, to maintain a dynamic equilibrium and avoid task dominating, losses associated with different tasks are scaled to the same constant. A straightforward way is to set each loss weight to the reciprocal of the corresponding loss, i.e., $w_i = \text{sg}(\frac{1}{\mathcal{L}_i}), i \in \{o, r, d\}$, where sg is a stop gradient function. However, as the loss is only calculated from a mini-batch of data and

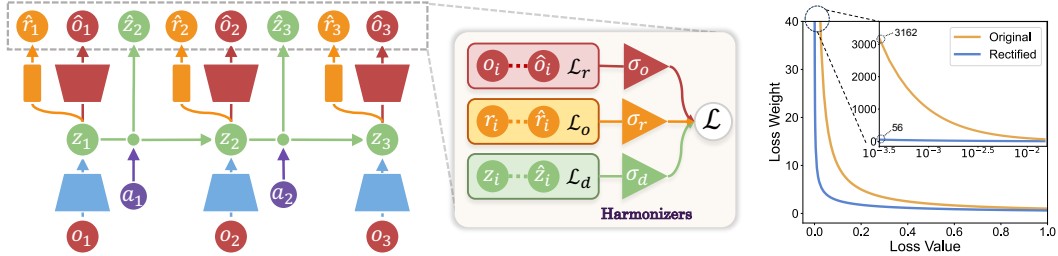

Figure 6: Overview of Harmony World Models. (*Left*) Built upon Dreamer, we introduce lightweight harmonizers to maintain a dynamic equilibrium between tasks. (*Right*) Comparison between the original harmonious loss and the rectified one. The latter prevents extremely large loss weights.

fluctuates throughout training, these weights are sensitive to outlier values and thus may further aggravate training instability. Instead, we adopt a variational method to learn the weights of different losses by the following *harmonious loss* for world model learning:

$$\mathcal{L}(\theta, \sigma_o, \sigma_r, \sigma_d) = \sum_{i \in \{o,r,d\}} \mathcal{H}(\mathcal{L}_i(\theta), \sigma_i) = \sum_{i \in \{o,r,d\}} \frac{1}{\sigma_i} \mathcal{L}_i(\theta) + \log \sigma_i. \tag{4}$$

The variational formulation $\mathcal{H}(\mathcal{L}_i(\theta), \sigma_i) = \sigma_i^{-1} \mathcal{L}_i(\theta) + \log \sigma_i$ serves as *harmonizers* to dynamically but smoothly rescale different losses, where the weight $\sigma_i^{-1}$ with a learnable parameter $\sigma_i > 0$ approximates a "global" reciprocal of the loss scale, as stated in the following proposition:

**Proposition 3.1.** *The optimal solution $\sigma^*$ that minimizes the expected loss $\mathbb{E}[\mathcal{H}(\mathcal{L}, \sigma)]$, or equivalently $\nabla_\sigma \mathbb{E}[\mathcal{H}(\mathcal{L}, \sigma)] = 0$, is $\sigma^* = \mathbb{E}[\mathcal{L}]$. In other words, the harmonized loss scale is $\mathbb{E}[\mathcal{L}/\sigma^*] = 1$.*

In practice, $\sigma_i$ is parameterized as $\sigma_i = \exp(s_i) > 0$, in order to optimize parameters $s_i$ free of sign constraint. More essentially, we propose a rectification on Eq. (4), as a loss $\mathcal{L}$ with small values, such as the reward loss, can lead to extremely large coefficient $1/\sigma \approx \mathcal{L}^{-1} \gg 1$, which potentially hurt training stability. Specifically, we simply add a constant in regularization terms:

$$\mathcal{L}(\theta, \sigma_o, \sigma_r, \sigma_d) = \sum_{i \in \{o,r,d\}} \hat{\mathcal{H}}(\mathcal{L}_i, \sigma_i) = \sum_{i \in \{o,r,d\}} \frac{1}{\sigma_i} \mathcal{L}_i(\theta) + \log(1 + \sigma_i). \tag{5}$$

The harmonized loss scale by the *rectified harmonious loss* is equal to $\frac{2}{1 + \sqrt{1 + 4/\mathbb{E}[\mathcal{L}]}} < 1$ (derivations in Appendix B), and we illustrate the corresponding loss weights learned with different loss scales in the right of Fig. 6, showing that the rectified loss effectively mitigates extremely large loss weights.

**Discussion.** Our harmonious loss takes a similar form as uncertainty weighting (Kendall et al., 2018) but has several key differences. Uncertainty weighting is derived from maximum likelihood estimation, which parameterizes noises of Gaussian-distributed outputs of each task, known as homoscedastic uncertainty. In contrast, our motivation is to harmonize loss scales among tasks. More specifically, measuring the uncertainty of observations and rewards results in putting each observation pixel on equal footing as the scalar reward, still overlooking the large disparity in dimension sizes. However, we take high-dimensional observations as a whole and directly balance the two losses. Furthermore, we do not make assumptions on the distributions behind losses, which makes it possible for us to balance the KL loss, while uncertainty weighting has no theoretical basis for doing so.

## 4 EXPERIMENTS

We evaluate the ability of HarmonyWM to boost sample efficiency of base MBRL methods on three visual control domains: Meta-world (Sec. 4.1, Yu et al. (2020b)), RLBench (Sec. 4.2, James et al. (2020)), and DMC Remastered (Sec. 4.3, Grigsby & Qi (2020)). These benchmarks contain diversified and challenging visual robotics manipulation and locomotion tasks. We conduct most of our experiments for HarmonyWM based on DreamerV2 but also demonstrate its generality to other base MBRL methods, including DreamerV3 (Hafner et al., 2023) and DreamerPro (Deng et al., 2022) (Sec. 4.4). Following Agarwal et al. (2021) and Seo et al. (2022a), we report mean values with 95% confidence intervals (CI) across 5 individual runs for each task. Experimental details and additional results can be found in Appendix C and D, respectively.

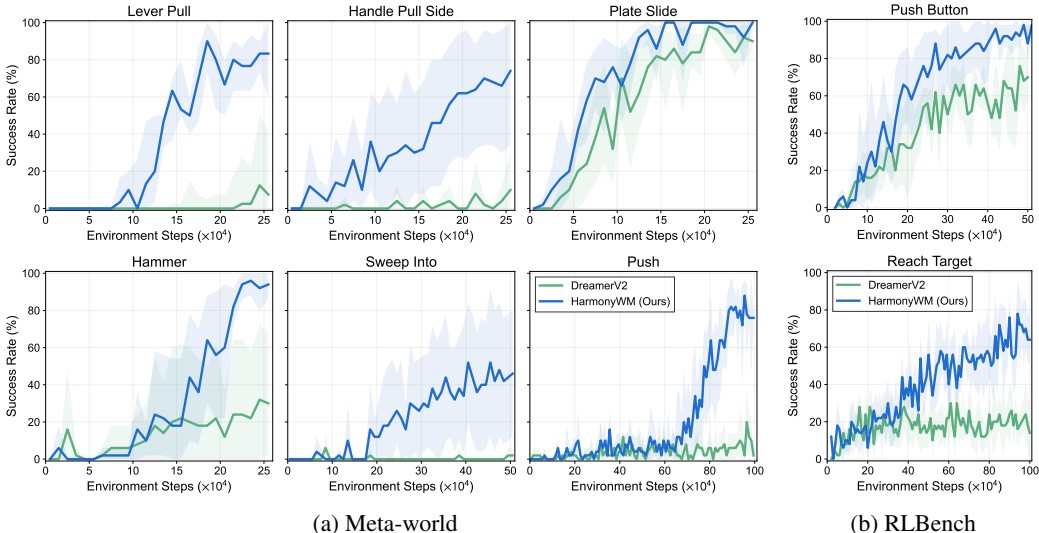

Figure 7: Learning curves on visual manipulation tasks from the (a) Meta-World and (b) RLBench benchmarks, measured on the success rate. We report the mean and 95% CI across five runs.

## 4.1 META-WORLD EXPERIMENTS

**Environment details.** Meta-world is a benchmark consisting of 50 distinct robotic manipulation tasks. Seo et al. (2022a) classify these tasks into four categories according to the task difficulty: *easy*, *medium*, *hard*, and *very hard*. Due to our limited computational resources, we choose a set of representative tasks: three from the *easy* category (Handle Pull Side, Lever Pull, and Plate Slide), two from the *medium* category (Hammer and Sweep Into), and one from the *hard* category (Push). These tasks are run over different numbers of environment steps: *easy* tasks and Hammer over 250K steps, Sweep Into over 500K steps, and Push over 1M steps.

**Results.** In Fig. 7a, we report the performance of HarmonyWM on six Meta-world tasks, in comparison with our base MBRL method DreamerV2. By simply adding harmonizers to the original DreamerV2 method, our HarmonyWM demonstrates superior performance in terms of both sample efficiency and final success rate. In particular, HarmonyWM achieves over 75% success rate on the challenging Push task, while DreamerV2 fails to learn a meaningful policy. For the Plate Slide task, where DreamerV2 is able to achieve a high success rate in 100K environment steps, we observe that HarmonyWM does not harm the learning process and still provides benefits.

## 4.2 RLBENCH EXPERIMENTS

**Environment details.** To assess our method on more complex visual robotic manipulation tasks, we perform further evaluation on the RLBench (James et al., 2020) domain. Most tasks in RLBench have high intrinsic difficulty and only offer sparse rewards. Learning these tasks requires expert demonstrations, dedicated network structure, and additional inputs (James & Davison, 2022; James et al., 2022), which is overchallenging for DreamerV2, even equipped with our powerful HarmonyWM. Therefore, following Seo et al. (2022a), we conduct experiments on two relatively easy tasks (Push Button and Reach Target) with dense rewards.

**Results.** In Fig. 7b, we show the superiority of our approach on the RLBench domain. HarmonyWM offers 28% of **absolute** final performance gain on the Push Button task and 50% on the more difficult Reach Target tasks. The results presented above prove the ability of HarmonyWM to promote sample efficiency of model-based RL on robotics manipulation domains for easy and difficult tasks alike.

## 4.3 DMC REMASTERED EXPERIMENTS

**Environment details.** DMC Remastered (DMCR, Grigsby & Qi (2020)) is a challenging extension of the widely used robotics control benchmark, DeepMind Control Suite (Tassa et al., 2018) with

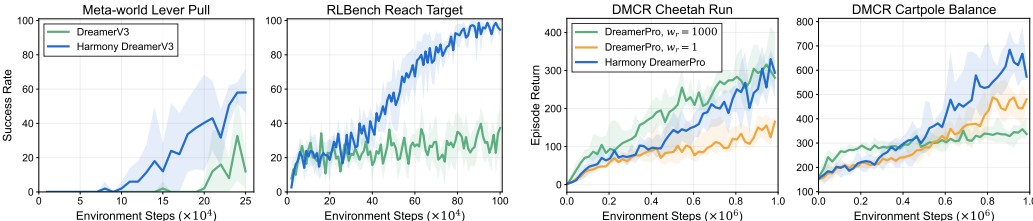

Figure 8: Example observations (*left*) and learning curves (*right*) on three DMC Remastered visual locomotion task. We report the mean and 95% CI across five runs, measured on the episode return.

Figure 9: Performance of HarmonyWM generalized to DreamerV3 (*left*) and DreamerPro (*right*).

randomly generated graphics emphasizing visual diversity, as shown in the left of Fig. 8. We train and evaluate our agents on three tasks: Cheetah Run, Walker Run, and Cartpole Balance.

**Results.** Fig. 8 demonstrates the effectiveness of HarmonyWM on three DMCR tasks. Our method greatly enhances the base DreamerV2 method to unleash its potential. Fig. 10a shows different learning curves of the dynamics loss between HarmonyWM and DreamerV2 on the DMCR domain. It is worth noting that DMCR tasks contain distracting visual factors, such as background and robot body color, which may hinder the learning process of observation modeling. By leveraging the importance of reward modeling, the model bypasses distractors in observations and can learn task-centric transitions with lower difficulty, indicated by converged dynamic loss. Through achieving outstanding performance on the DMCR domain, we show that HarmonyWM consistently improves the sample efficiency of model-based RL in the field of robotic locomotion.

## 4.4 ANALYSIS

**Comparisons to implicit MBRL.** As we have shown in Sec. 2.3, learning from reward modeling alone lacks in capability for sample-efficient learning. However, as DreamerV2 is intentionally built as an explicit MBRL method, one may argue that purposefully designed implicit MBRL methods can be more effective. In Fig. 10b, we show comparisons with a prevalent implicit MBRL method, TD-MPC (Hansen et al., 2022) on three tasks of Meta-world. We observe that TD-MPC has difficulty in efficient learning as it lacks observation modeling to guide representation learning. In contrast, our method dynamically balances reward modeling and observation modeling, as shown in Fig. 10c and achieves superior performance, therefore proving the value of our *explicit-implicit* MBRL method.

**Method generality.** DreamerV3 (Hafner et al., 2023) builds upon DreamerV2 and is a general algorithm that learns to master diverse domains while using fixed hyperparameters. Experiments with DreamerV3 are done on the Meta-world and RLBench domains, where we choose a typical task from each domain. DreamerPro (Deng et al., 2022) is a reconstruction-free model-based RL method that "reconstructs" the cluster assignment of the observation instead of the observation itself. We conduct DreamerPro experiments on the DMCR benchmark, given that DreamerPro has shown outstanding performance in natural background DMC, a setting similar to DMCR. Implementation details are listed in Appendix C.2. By default, DreamerPro uses a manually tuned reward loss weight $w_r = 1000$. We show in Fig. 9 that by generalizing HarmonyWM to other base MRBL methods, our method can also achieve higher sample efficiency and, on average, outperform manually tuned weights that are computationally costly. These results show the consistent effectiveness and excellent generality of HarmonyWM on base MBRL methods. We provide more experimental results in Appendix D.

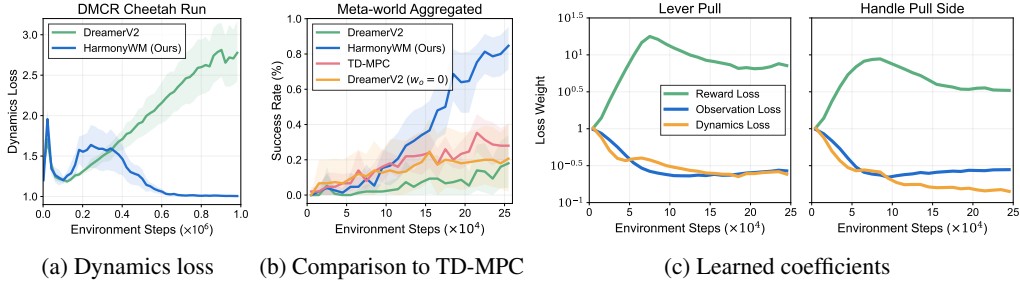

(a) Dynamics loss     (b) Comparison to TD-MPC     (c) Learned coefficients

Figure 10: Analysis of HarmonyWM and comparison to implicit MBRL method.

## 5 RELATED WORK

**World models for visual RL.** There exist several approaches to learning world models that explicitly model observations, transitions, and rewards. They can be widely utilized to boost sample efficiency in visual RL. In world models, visual representation can be learned via image reconstruction (Ha & Schmidhuber, 2018; Kaiser et al., 2020; Zhang et al., 2019; Hafner et al., 2019; Seo et al., 2022a;b; Robine et al., 2023), or reconstruction-free contrastive learning (Okada & Taniguchi, 2021; Deng et al., 2022). Dreamer (Hafner et al., 2020; 2021; 2023) represents a series of methods that learn latent dynamics models from observations and learn behaviors by latent imagination. These methods have proven their effectiveness in tasks such as video games (Hafner et al., 2021) and visual robot control (Wu et al., 2022). Regardless, the problem of task domination is general for world models, and our findings and approach are not limited to the Dreamer architecture.

**Implicit model-based RL.** Implicit MBRL (Oh et al., 2017; Hansen et al., 2022; Moerland et al., 2023) adopts a more abstract approach, viewing RL as a form of reward optimization, and aims to learn value equivalence models (Grimm et al., 2020) that focuses on task-centric characteristics of the environment. This approach mitigates the objective mismatch (Lambert et al., 2020) between maximum likelihood estimation for world models and maximizing returns for policies. A typical success is MuZero (Schrittwieser et al., 2020; Ye et al., 2021), which learns a world model by predicting task-specific rewards, values, and policies, without explicit reconstruction of complex observations. Our analysis shows that the potential efficiency of task-centric models can be better released when properly balanced with richer information from observation models.

**Multi-task learning.** Multi-task learning (Caruana, 1997; Ruder, 2017) aims to improve the performance of different tasks by jointly learning from a shared representation. The common approach is to aggregate task losses, where the loss or gradient of each task is manipulated by criteria such as uncertainty (Kendall et al., 2018), performance metrics (Guo et al., 2018), gradient norm (Chen et al., 2018) or gradient direction (Yu et al., 2020a; Wang et al., 2021; Navon et al., 2022), to avoid negative transfer (Jiang et al., 2023). Previous works on multi-task learning in RL typically considered different tasks of policy learning defined by different reward functions or environment dynamics (Rusu et al., 2016; Teh et al., 2017; Yu et al., 2020a). In contrast to all prior work, we depict world model learning as multi-task learning, composed of reward and observation modeling, and our HarmonyWM learns to maintain a delicate equilibrium between them to mitigate domination.

## 6 CONCLUSION

We adopt a multi-task perspective into world models, unifying explicit and implicit MBRL through different task weighting. Our empirical study reveals an oversight in prior literature: domination of a particular task can dramatically deteriorate the sample efficiency of MBRL. We thus introduce the Harmony World Model, a simple approach designed to dynamically balance these tasks, substantially improving sample efficiency. To improve our work, we aim to extend HarmonyWM to better harmonize more complex objectives (Hafner et al., 2023) in world models beyond simply considering loss scales. More intricate task hierarchies should also be investigated. For instance, sub-tasks exist in observation modeling, including latent transitions and observation decoding, while the latter may also involve multimodal signals (Wu et al., 2022). Overall, we hope our work can provide insights and help pave the path ahead for exploring and exploiting the multi-task nature of world models.

## REPRODUCIBILITY STATEMENT

To ensure the reproducibility and completeness of this paper, we include the Appendix with four main sections. Appendix A explains the process of actor-critic learning for our HarmonyWM. Derivations of the mechanism behind our harmonious losses are shown in Appendix B. The experiments in the paper are reproducible with additional implementation details provided in Appendix C. We also include the hyperparameter settings for all results reported in Fig. 7 and 8. Appendix D contains additional experiment results. Code will be made available upon publication.

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

## A  BEHAVIOR LEARNING

Our HarmonyWM does not change the behavior learning procedure of its base MBRL methods (Hafner et al., 2021; 2023; Deng et al., 2022), and we briefly describe the actor-critic learning scheme shared by these base methods and HarmonyWM.

Specifically, we leverage a stochastic actor and a deterministic critic parameterized by $\psi$ and $\xi$, respectively, as shown below:

$$\text{Actor: } \hat{a}_t \sim \pi_\psi\left(\hat{a}_t \mid \hat{z}_t\right) \quad \text{Critic: } v_\xi\left(\hat{z}_t\right) \approx \mathrm{E}_{p_\theta, \pi_\psi}\left[\sum\nolimits_{\tau \geq t} \gamma^{\tau-t} \hat{r}_\tau\right], \tag{6}$$

where $p_\theta$ is the world model. The actor and critic are jointly trained on the same imagined trajectories $\{\hat{z}_\tau, \hat{a}_\tau, \hat{r}_\tau\}$ with horizon $H$, generated by the transition model and reward model in Eq. (1) and the actor in Eq. (6). The critic is trained to regress the $\lambda$-target:

$$\mathcal{L}_{\text{critic}}(\xi) \doteq \mathbb{E}_{p_\theta, \pi_\psi}\left[\sum_{\tau=t}^{t+H} \frac{1}{2}\left(v_\xi(\hat{z}_\tau) - \text{sg}(V_\tau^\lambda)\right)^2\right], \tag{7}$$

$$V_\tau^\lambda \doteq \hat{r}_\tau + \gamma \begin{cases}(1-\lambda)v_\xi(\hat{z}_{\tau+1}) + \lambda V_{\tau+1}^\lambda & \text{if } \tau < t + H \\ v_\xi(\hat{z}_{\tau+1}) & \text{if } \tau = t + H.\end{cases} \tag{8}$$

The actor, meanwhile, is trained to output actions that maximize the critic output by backpropagating value gradients through the learned world model. The actor loss is defined as follows:

$$\mathcal{L}_{\text{actor}}(\psi) \doteq \mathbb{E}_{p_\theta, \pi_\psi}\left[\sum_{\tau=t}^{t+H}\left(-V_\tau^\lambda - \eta\, \mathrm{H}\left[\pi_\psi(\hat{a}_\tau|\hat{z}_\tau)\right]\right)\right], \tag{9}$$

where $\mathrm{H}\left[\pi_\psi(\hat{a}_\tau|\hat{z}_\tau)\right]$ is an entropy regularization which encourages exploration, and $\eta$ is the hyperparameter that adjusts the regularization strength. For more details, we refer to Hafner et al. (2020).

## B  DERIVATIONS

**Proof of Proposition 3.1.**  To minimize $\mathbb{E}[\mathcal{H}(\mathcal{L}, \sigma)]$, we force the the partial derivative w.r.t. $\sigma$ to 0:

$$\nabla_\sigma \mathbb{E}[\mathcal{H}(\mathcal{L}, \sigma)] = \nabla_\sigma \mathbb{E}\left[\frac{1}{\sigma}\mathcal{L} + \log \sigma\right] = \mathbb{E}\left[\nabla_\sigma\left(\frac{1}{\sigma}\mathcal{L} + \log \sigma\right)\right] \tag{10}$$

$$= \mathbb{E}\left[-\frac{1}{\sigma^2}\mathcal{L} + \frac{1}{\sigma}\right] = \frac{1}{\sigma} - \frac{1}{\sigma^2}\mathbb{E}[\mathcal{L}] = 0. \tag{11}$$

This results in the solution $\sigma^* = \mathbb{E}[\mathcal{L}]$, and equivalently, the harmonized loss scale is $\mathbb{E}[\mathcal{L}/\sigma^*] = 1$.

**Analytic solution of rectified loss.**  Similarly, minimizing $\mathbb{E}\left[\hat{\mathcal{H}}(\mathcal{L}, \sigma)\right]$ yields

$$\nabla_\sigma \mathbb{E}\left[\hat{\mathcal{H}}(\mathcal{L}, \sigma)\right] = \nabla_\sigma\left(\frac{1}{\sigma}\mathbb{E}[\mathcal{L}] + \log(1 + \sigma)\right) = -\frac{1}{\sigma^2}\mathbb{E}[\mathcal{L}] + \frac{1}{1+\sigma} = 0$$

$$\sigma = \frac{\mathbb{E}[\mathcal{L}] + \sqrt{\mathbb{E}[\mathcal{L}]^2 + 4\mathbb{E}[\mathcal{L}]}}{2}. \tag{12}$$

Therefore the learnable loss weight, in our rectified harmonious loss, approximates the analytic loss weight:

$$\frac{1}{\sigma} = \frac{2}{\mathbb{E}[\mathcal{L}] + \sqrt{\mathbb{E}[\mathcal{L}]^2 + 4\mathbb{E}[\mathcal{L}]}}, \tag{13}$$

corresponding to a loss scale $\mathbb{E}[\mathcal{L}]$, which is less than the unrectified $1/\mathbb{E}[\mathcal{L}]$. Adding a constant in the regularization term $\log(1 + \sigma)$ results in the $4\mathbb{E}[\mathcal{L}]$ in the $\sqrt{\mathbb{E}[\mathcal{L}]^2 + 4\mathbb{E}[\mathcal{L}]}$ term, which prevents the loss weight from getting extremely large when faced with a small $\mathbb{E}[\mathcal{L}]$.

## C  EXPERIMENTAL DETAILS

### C.1  BENCHMARK ENVIRONMENTS

**Meta-world.**  Meta-world (Yu et al., 2020b) is a benchmark of 50 distinct robotic manipulation tasks. We choose six tasks in all according to the difficulty criterion *(easy, medium, hard, and very hard)* proposed by Seo et al. (2022a). Specifically, we choose Handle Pull Side, Lever Pull, and Plate Slide from the *easy* category, Hammer and Sweep Into from the *medium* category, and Push from the *hard* category. We observe that although the Hammer task belongs to the *medium* category, it is relatively easy for the DreamerV2 agent to learn, and our HarmonyWM can already achieve high success by 250K environment steps. Therefore, we train our agents over 250K environment steps on Hammer, along with the three *easy* tasks. For the Sweep Into and Push task, we train our agents over 500K and 1M environment steps, according to their various difficulties, respectively. In all tasks, the episode length is 500 environment steps with no action repeat.

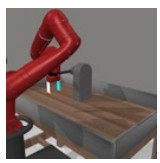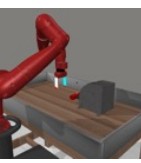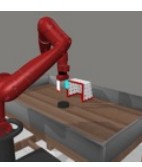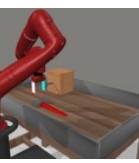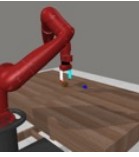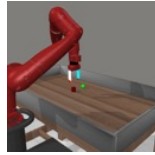

Figure 11: Example observations of Meta-world tasks. (From left to right: Lever Pull, Handle Pull Side, Plate Slide, Hammer, Sweep Into, Push)

**RLBench.**  RLBench (James et al., 2020) is a challenging benchmark for robot learning. Most tasks in RLBench are overchallenging for DreamerV2, even equipped with HarmonyWM. Therefore, following Seo et al. (2022a), we choose two relatively easy tasks (i.e. Push Button, Reach Target) and use an action mode that specifies the delta of joint positions. Because the original RLBench benchmark does not provide dense rewards for the Push Button task, we assign a dense reward following Seo et al. (2022a), which is defined as the sum of the L2 distance of the gripper to the button and the magnitude of the button being pushed. In our experiments, We found that the original convolutional encoder and decoder of DreamerV2 can be insufficient for learning the RLBench task. Therefore, in this domain, we adopt the ResNet-style encoder and decoder from Wu et al. (2023) for both DreamerV2 and our HarmonyWM. Note here that changes in the encoder and decoder architecture are completely orthogonal to our method and contributions. For tasks in the RLBench domain, the maximum episode length is set to 400 environment steps with an action repeat of 2.

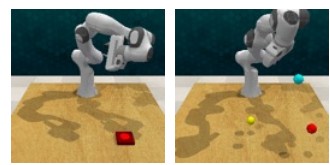

Figure 12: Example observations of RL-Bench tasks: Push Button and Reach Target.

**DMC Remastered.**  The DMC Remastered (DMCR) (Grigsby & Qi, 2020) benchmark is a challenging extension of the widely used robotic locomotion benchmark, DeepMind Control Suite (Tassa et al., 2018), by expanding a complicated graphical variety. On initialization of each episode for both training and evaluation, the DMCR environment randomly resets 7 factors affecting visual conditions, including floor texture, background, robot body color, target color, reflectance, camera position, and lighting. Our agents are trained and evaluated on three tasks: Cheetah Run, Walker Run, and Cartpole Balance. We use all variation factors in all of our experiments and train our agents over 1M environment steps. Following the common setup of DeepMind Control Suite (Hafner et al., 2020; Yarats et al., 2022), we set the episode length to 1000 environment steps with an action repeat of 2.

### C.2  BASE MBRL METHODS

**DreamerV2.**  Our HarmonyWM is built upon DreamerV2 (Hafner et al., 2021), which has been elaborated on in the main text, and we refer the readers to Sec. 2.2 and Hafner et al. (2020; 2021).

**DreamerV3.** DreamerV3 (Hafner et al., 2023) is a general and scalable algorithm that builds upon DreamerV2. In order to master a wide range of domains with fixed hyperparameters, DreamerV3 made many changes to DreamerV2, including using symlog predictions, utilizing world model regularization by combining KL balancing and free bits, modifying the network architecture, and so forth. A main modification relevant to our method is that DreamerV3 explicitly partitions the dynamics loss in Eq. (2) into a dynamics loss and a representation loss as follows:

Dynamics loss: $\quad \mathcal{L}_{\mathrm{dyn}}(\theta) = \max(1, \mathrm{KL}\left[\mathrm{sg}(q_\theta(z_t \mid z_{t-1}, a_{t-1}, o_t)) \,\|\, p_\theta(\hat{z}_t \mid z_{t-1}, a_{t-1})\right]),$

Representation loss: $\quad \mathcal{L}_{\mathrm{rep}}(\theta) = \max(1, \mathrm{KL}\left[q_\theta(z_t \mid z_{t-1}, a_{t-1}, o_t) \,\|\, \mathrm{sg}(p_\theta(\hat{z}_t \mid z_{t-1}, a_{t-1}))\right]).$

(14)

Since $\mathcal{L}_{\mathrm{dyn}}(\theta)$ and $\mathcal{L}_{\mathrm{rep}}(\theta)$ is of the same loss value, which will result in same learned coefficients, to implement Harmony DreamerV3, we recombine the two losses into $\mathcal{L}_d(\theta)$ as follows:

$$\mathcal{L}_d(\theta) \doteq \mathcal{L}_{\mathrm{dyn}}(\theta) + \mathcal{L}_{\mathrm{rep}}(\theta). \tag{15}$$

In this way, we can use the same learning objective as Eq. (5) for Harmony DreamerV3.

**DreamerPro.** DreamerPro (Deng et al., 2022) is a reconstruction-free model-based RL method that incorporates prototypical representations in the world model learning process. The overall learning objective of the DreamerPro method is defined as follows:

$$\mathcal{L}_{\mathrm{DreamerPro}}(\theta) = \mathcal{L}_{\mathrm{SwAV}}(\theta) + \mathcal{L}_{\mathrm{Temp}}(\theta) + \mathcal{L}_{\mathrm{R}}(\theta) + \mathcal{L}_{\mathrm{KL}}(\theta). \tag{16}$$

The $\mathcal{L}_{\mathrm{SwAV}}$ term stands for prototypical representation loss used in SwAV (Caron et al., 2021), which improves prediction from an augmented view and induces useful features for static images. $\mathcal{L}_{\mathrm{Temp}}$ stands for temporal loss that considers temporal structure and reconstructs the cluster assignment of the observation instead of the visual observation itself. As $\mathcal{L}_{\mathrm{SwAV}} + \mathcal{L}_{\mathrm{Temp}}$ replaces $\mathcal{L}_o$ in Eq. (2), we build our Harmony DreamerPro by substituting the overall learning objective into the following:

$$\mathcal{L}_{\mathrm{Harmony\ DreamerPro}}(\theta) = \sum_{i \in \{\mathrm{SwAV,Temp,R,KL}\}} \frac{1}{\sigma_i} \mathcal{L}_i(\theta) + \log\left(1 + \sigma_i\right). \tag{17}$$

### C.3 HYPERPARAMETERS

Our proposed HarmonyWM only involves adding lightweight harmonizers, each corresponding to a single learnable parameter, and thus does not introduce any additional hyperparameters. For Harmony DreamerV3 and Harmony DreamerPro, we use the default hyperparameters of DreamerV3 and DreamerPro, respectively. For our HarmonyWM, we use the same set of hyperparameters as DreamerV2 (Hafner et al., 2021). Important hyperparameters for HarmonyWM are listed in Table 1.

### C.4 ANALYSIS EXPERIMENT DETAILS (FIG. 4 AND 5)

For the analysis in Sec. 2.3, namely Fig. 4 and 5, we conduct our experiments on a fixed training buffer to better ablate distracting factors. We first train a separate DreamerV2 agent and use training trajectories collected during its whole training process as our fixed buffer. The fixed buffer comprises 250K environment steps and covers data from low-quality to high-quality trajectories (Levine et al., 2020). We then offline train our DreamerV2 agents with different reward loss coefficients on this buffer. All other hyperparameters, such as training frequency, training steps, and evaluation episodes, are the same as in Table 1.

**Details for Fig. 4** We denote the agent trained using $w_r = 1$ as *original weight* and trained using $w_r = 100$ for Lever Pull, $w_r = 10$ for Handle Pull Side and Hammer as *balanced weight*. To build the state regression dataset, first, we gather 10,000 segments of trajectories, each with a length of 50, from the evaluation episodes of both the agent trained using *original weight* and the agent trained using *balanced weight*. These segments are then combined into a dataset comprising 20,000 segments. This dataset is subsequently divided into a training set and a validation set at a ratio of 90% to 10%, respectively. Each data point in the dataset consists of a ground truth state and a predicted state representation, where the ground truth state is made up of the actual positions of task-relevant objects. We use a 4-layer MLP with a hidden size of 400 and an MSE loss to regress the representation to the ground-truth state. We report regression loss results on the validation set.

Table 1: Hyperparameters in our experiments. We use the same hyperparameters as DreamerV2.

| Hyperparameter | Value |
|---|---|
| Observation size | $64 \times 64 \times 3$ |
| Observation preprocess | Linearly rescale from $[0, 255]$ to $[-0.5, 0.5]$ |
| Action Repeat | 1 for Meta-world |
| | 2 for RLBench and DMCR |
| Max episode length | 500 for Meta-world and DMCR, 200 for RLBench |
| Early episode termination | True for RLBench, False otherwise |
| Trajectory segment length $T$ | 50 |
| Random exploration | 5000 environment steps for Meta-world and RLBench |
| | 1000 environment steps for DMCR |
| Replay buffer capacity | $10^6$ |
| Training frequency | Every 5 environment steps |
| Imagination horizon $H$ | 15 |
| Discount $\gamma$ | 0.99 |
| $\lambda$-target discount | 0.95 |
| Entropy regularization $\eta$ | $1 \times 10^{-4}$ |
| Batch size | 50 for Meta-world and RLBench |
| | 16 for DMCR |
| RSSM hidden size | 1024 |
| World model optimizer | Adam |
| World model learning rate | $3 \times 10^{-4}$ |
| Actor optimizer | Adam |
| Actor learning rate | $8 \times 10^{-5}$ |
| Critic optimizer | Adam |
| Critic learning rate | $8 \times 10^{-5}$ |
| Evaluation episodes | 10 |

**Details for Fig. 5** In the Lever Pull task, the robot needs to reach the end of a lever (marked in blue in the observation) and pull it to the designated position (marked in red in the observation). We utilize a trajectory where the default DreamerV2 with $w_r = 1$ fails to lift the lever to analyze the reason behind its poor performance. Both agents use 15 frames for observation and reconstruction and predict 35 frames open-loop. We plot each image with an interval of 5 frames in Fig. 5.

### C.5 COMPUTATIONAL RESOURCES

We implement our HarmonyWM based on PyTorch (Paszke et al., 2019). Training is conducted with automatic mixed precision (Micikevicius et al., 2018) on Meta-world and RLBench and full precision on DMCR. In terms of training time, it takes ~24 hours for each run of Meta-world experiments over 250K environment steps, ~24 hours for RLBench over 500K environment steps, and ~23 hours for DMCR over 1M environment steps, respectively. The lightweight harmonizers introduced by HarmonyWM do not affect the training time. In terms of memory usage, Meta-world and RLBench experiments require ~10GB GPU memory, and DMCR requires ~5GB GPU memory, thus, the experiments can be done using typical 12GB GPUs.

## D  ADDITIONAL EXPERIMENT RESULTS

### D.1  EXPERIMENTAL SUPPORT ON RECTIFIED HARMONIOUS LOSS

In Sec. 3, we have already presented a detailed explanation on the necessity of our *rectified harmonious loss*, changing the regularization term from $\log \sigma_i$ in Eq. (4) to $\log(1 + \sigma_i)$ in Eq. (5). Here, we present experimental results to support our claim. We use *Unrectified HarmonyWM* to note our method trained using the objective in Eq. (4), and *HarmonyWM (Ours)* to note our method trained using Eq. (5). As shown in Fig. 13 and Fig. 14, the excessively large reward coefficient (Fig. 13c) for *Unrectified HarmonyWM* can lead to a divergence in the dynamics loss (Fig. 13b), which in turn negatively impacts performance (Fig. 13a and Fig. 14).

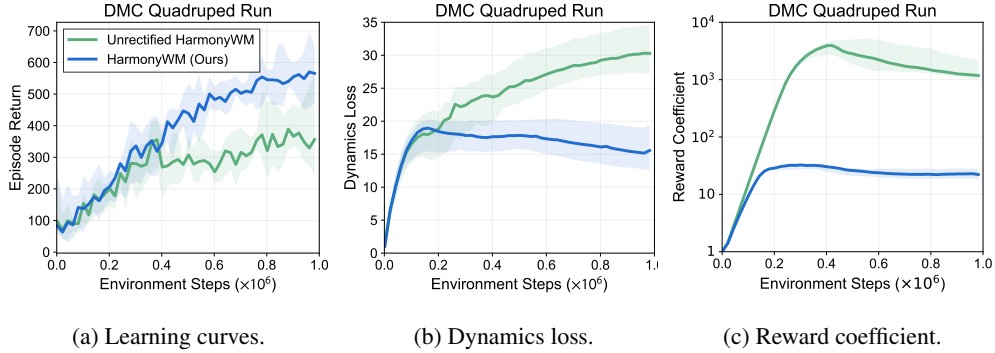

(a) Learning curves.      (b) Dynamics loss.      (c) Reward coefficient.

Figure 13: Training curves for *Unrectified HarmonyWM* using Eq. (4) on the DMC Quadruped Run task, in comparison with HarmonyWM.

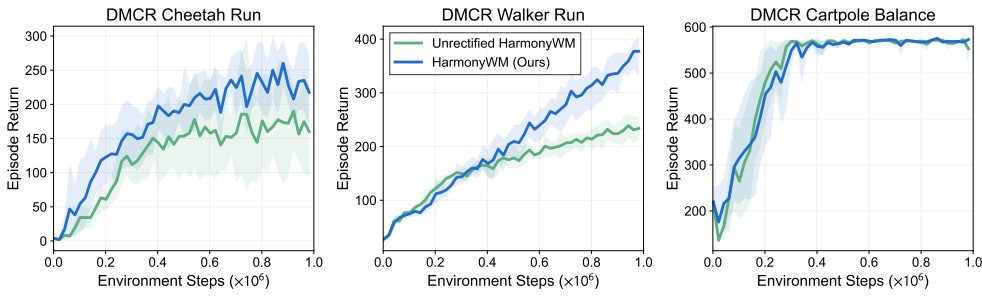

Figure 14: Learning curves for *Unrectified HarmonyWM* using Eq. (4) on the DMCR domain, in comparison with HarmonyWM.

## D.2 ABLATION ON ADJUSTING $w_d$

Manually tuning the dynamics loss coefficient $w_d$ (e.g. $w_d = 0.1$) is common in MBRL methods (Hafner et al., 2021; 2023; Seo et al., 2022a;b). We note that our HarmonyWM differs from these previous approaches as we treat the different losses in a multi-task perspective and harmonize loss scales between them, while previous approaches see $w_d$ simply as a hyperparameter. Fig. 15 shows a comparison between fixing $w_d$ to 1 in HarmonyWM (denoted as *HarmonyWM $w_d = 1$*) and using $\sigma_d$ to balance $w_d$ (denoted as *HarmonyWM (Ours)*), where our proposed HarmonyWM performs slightly better than the one fixing $w_d$, and both methods outperform DreamerV2 by a clear margin. This result highlights the importance of harmonizing two different modeling tasks in world models, instead of only tuning on the shared dynamics part of them.

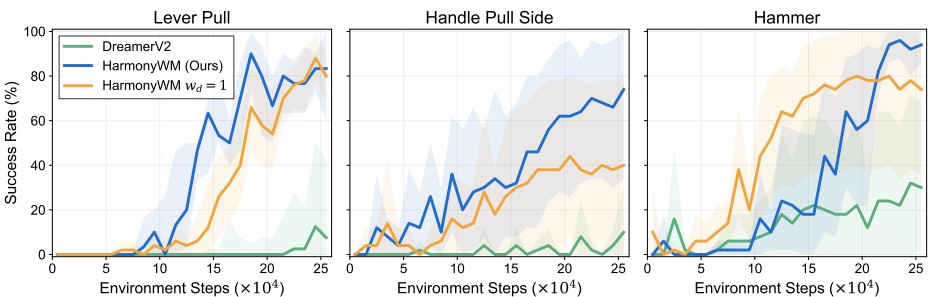

Figure 15: Ablation on adjusting $w_d$ in HarmonyWM.

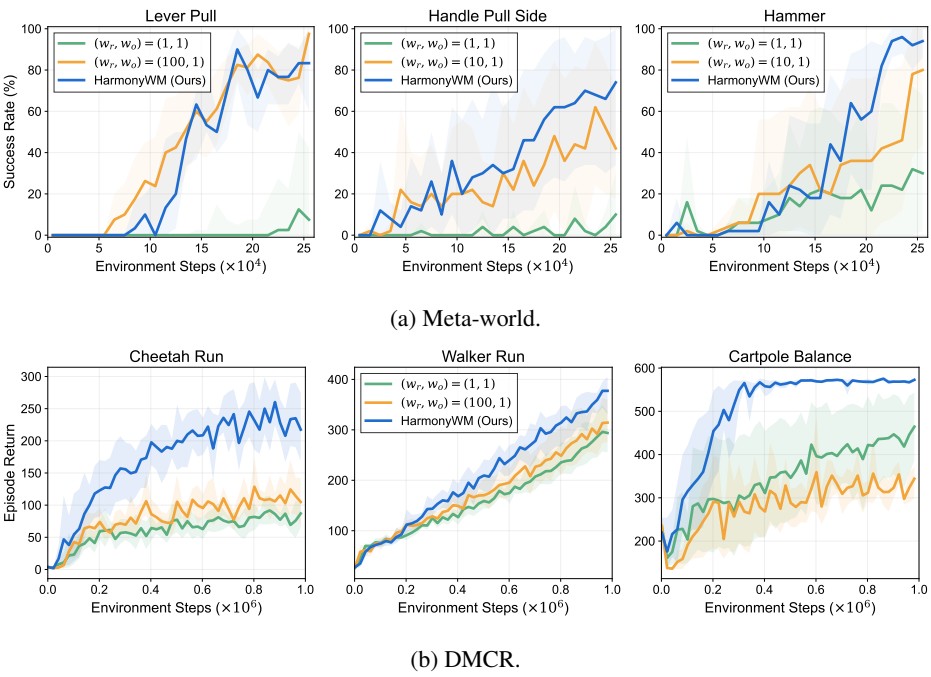

(a) Meta-world.

(b) DMCR.

Figure 16: Learning curves of HarmonyWM compared to tuned weights on Meta-world and DMCR.

### D.3 COMPARISON TO TUNED WEIGHTS

We present a direct comparison between our HarmonyWM and manually tuned weights for Dreamer-V2. For the Meta-world domain, we plot the tuned better results from $w_r \in \{10, 100\}, w_o = 1$. For the DMCR domain, we plot tuned results using $w_r = 100, w_o = 1$. Results in Fig. 16 show that our HarmonyWM outperforms manually tuned weights in most tasks, which adds to the value of our method.

### D.4 DEEPMIND CONTROL SUITE EXPERIMENTS

The DeepMind Control Suite (DMC, Tassa et al. (2018)) is a widely used benchmark for visual locomotion. We have conducted additional experiments on four tasks: Cheetah Run, Quadruped Run, Walker Run, and Finger Turn Hard. In Fig. 17, we present comparisons between our HarmonyWM and the base DreamerV2. We note that the performance of relatively easy DMC tasks has been almost saturated by recent literature (Yarats et al., 2021; Hafner et al., 2021), and we suppose that current limitations of model-based methods are not rooted in the world model, but rather in behavior learning (Hafner et al., 2023), which falls outside the scope of our method and contributions. Nevertheless, our HarmonyWM is still able to obtain a noticeable gain in performance in the more difficult Quadruped Run task.

### D.5 ADDITIONAL RESULTS OF IMPLICIT MBRL METHODS

We observe that the performance of TD-MPC (Hansen et al., 2022) is fairly low compared to our HarmonyWM. Due to our limited computational resources, we only conduct additional experiments on the DMCR domain, apart from the Meta-world results presented in Fig. 10b. Full results in Fig. 18 show that TD-MPC is unable to learn a meaningful policy in some tasks, which further highlights the value of our *explicit-implicit* method.

### D.6 ADDITIONAL RESULTS OF HARMONYWM

In Fig. 19, we present an additional result of our HarmonyWM on the **Assembly** task of the Meta-world domain. According to the difficulty criterion proposed by (Seo et al., 2022a), Assembly belongs

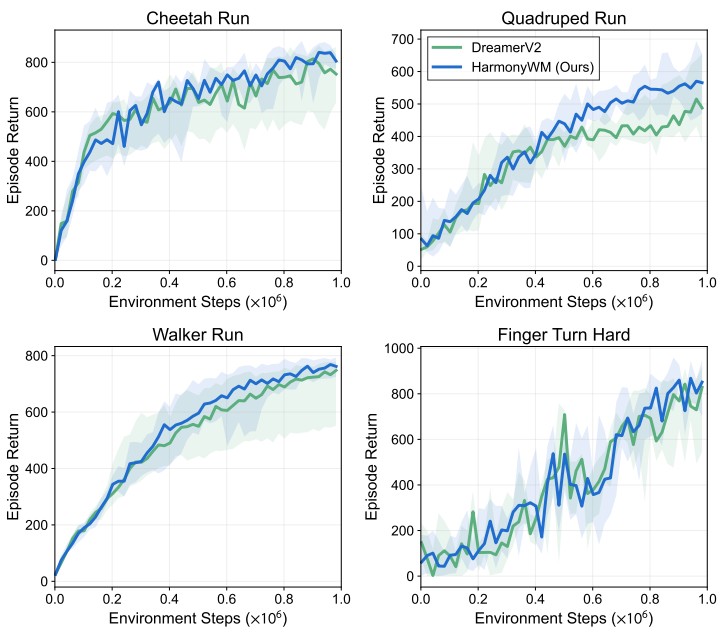

Figure 17: Learning curves of HarmonyWM and DreamerV2 on the DMC domain.

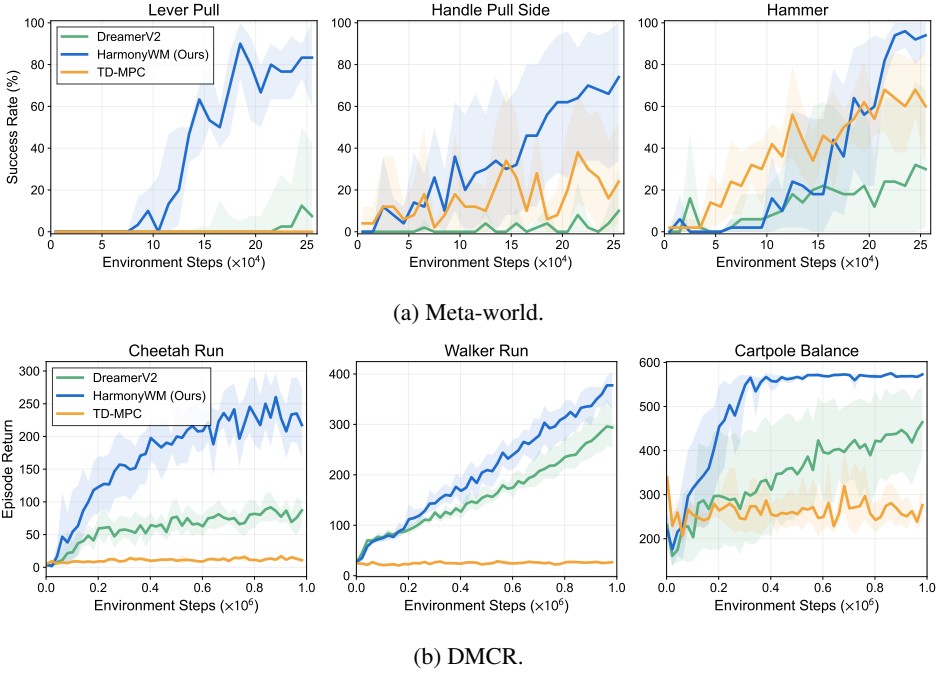

(a) Meta-world.

(b) DMCR.

Figure 18: Learning curves of TD-MPC.

to the *hard* category. We train our agents over 1M environment steps. This result further demonstrates the superiority of our method on challenging tasks.

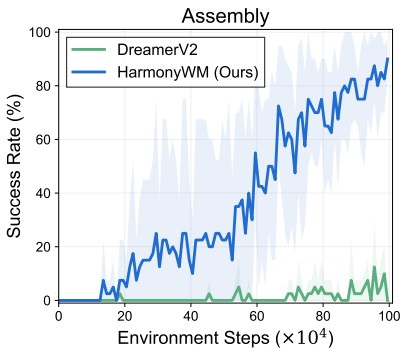

Figure 19: Additional Results of HarmonyWM on Assembly.

## D.7 ADDITIONAL RESULTS OF METHOD GENERALITY AND DISCUSSIONS

We present additional results of our HarmonyWM generalized to DreamerV3 in Fig. 20. We brand this method as Harmony DreamerV3 in the figure. Our approach consistently improves the sample efficiency of our base method, proving the excellent generality of our proposed harmonization.

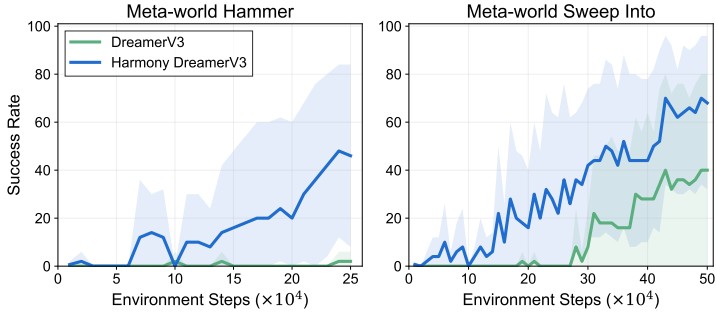

Figure 20: Additional results of Harmony DreamerV3 on Meta-world.

**Discussion** There are mainly two changes of DreamerV3 relevant to improving world model learning: KL Balancing and Symlog Predictions. We have already shown in Appendix C.2 that KL balancing is orthogonal to our method and that we can easily incorporate this modification into our approach. Besides, Symlog Predictions also do not solve our problem of seeking a balance between reward modeling and observation modeling. First of all, the Symlog transformation only shrinks extremely large values but is unable to rescale various values into exactly the same magnitude, while our harmonious loss properly addresses this by dynamically approximating the reciprocals of the values. More importantly, the primary reason why $L_r$ has a significantly smaller loss scale is the difference in dimension: as we have stated in Sec 2.3, the observation loss $L_o$ usually aggregates $H \times W \times C$ dimensions, while the reward loss $L_r$ is derived from only a scalar. In summary, using Symlog Predictions as DreamerV3 only mitigates the problem of differing per-dimension scales (typically across environment domains) by a static transformation, while our method aims to dynamically balance the overall loss scales across tasks in world model learning, considering together per-dimension scales, dimensions, and training dynamics.

In practice, DreamerV3 uses twohot symlog predictions for the reward predictor to replace the MSE loss in DreamerV2. This approach increases the scale of the reward loss, but is insufficient to mitigate the domination of the image loss. We observe that the reward loss in DreamerV3 is still significantly

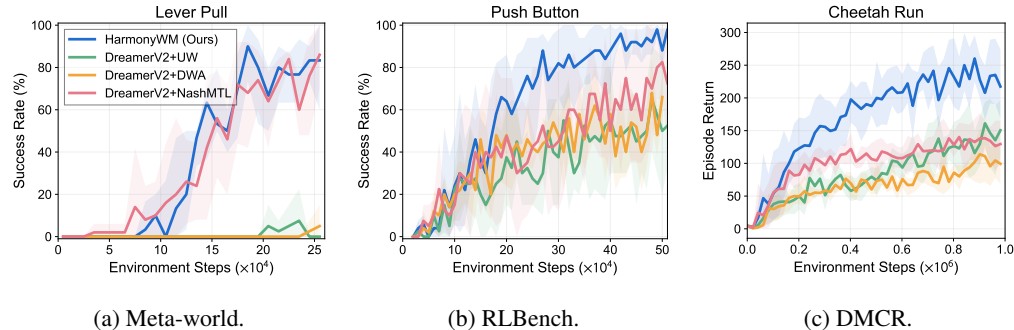

(a) Meta-world.          (b) RLBench.          (c) DMCR.

Figure 21: Comparison of HarmonyWM with multi-task learning methods.

smaller than the observation loss, especially for visually demanding domains such as RLBench, where the reward loss is still two orders of magnitude smaller.

## D.8 COMPARISON WITH MULTI-TASK LEARNING METHODS AND DISCUSSIONS

Methods in the field of multi-task learning or multi-objective learning can be roughly categorized into loss-based and gradient-based. Since gradient-based methods mainly address the problem of gradient conflicts (Yu et al., 2020a; Liu et al., 2021), which is not the main case in world model learning, we focus on loss-based methods, which assigns different weights to task losses by various criteria. We choose the following as our baselines to discuss differences and conduct comparison experiments:

**Uncertainty Weighting (UW, Kendall et al. (2018))** balances tasks with different scales of targets, which is measured as uncertainty of outputs. As pointed out in Section 2.2, in world model learning, observation loss $\mathcal{L}_o(\theta) = -\log p_\theta(o_t \mid z_t) = -\sum_{i=1}^{h}\sum_{j=1}^{w}\log p_\theta(o_t^{(i,j)} \mid z_t)$ and reward loss $\mathcal{L}_r(\theta) = -\log p_\theta(r_t \mid z_t)$ differs not only in scales but also in dimensions. A detailed explanation of the differences between our harmonious loss and UW is provided in the discussion section in Sec 3.

**Dynamics Weight Average (DWA, Liu et al. (2019))** balances tasks according to their learning progress, illustrating the various task difficulties. However, in world model learning, since the data in the replay buffer is growing and non-stationary, the relative descending rate of losses may not accurately measure task difficulties and learning progress.

**NashMTL (Navon et al., 2022)** is the most similar to our method, whose optimization direction has balanced projections of individual gradient directions. However, its implementation is far more complex than our method, as it introduces an optimization procedure to determine loss weights on each iteration. In our experiments, we also find this optimization is prone to deteriorate to produce near-zero weights without careful tuning of optimization parameters.

In Fig 21, we compare to the multi-task methods we mentioned above. Experiments are conducted on *Lever Pull* from Meta-world, *Push Button* from RLBench, and *Cheetah Run* from DMCR, respectively. Our method is the most effective among multi-task methods and has the advantage of simplicity. Although NashMTL produces similar results on the Lever Pull task, it outputs extreme task weights on the other two tasks, which accounts for its low performance. Our HarmonyWM, on the other hand, uses a rectified loss that effectively mitigates extremely large loss weights.

## D.9 COMPARISON WITH DENOISED MDP

Our HarmonyWM shares a similar point with Denoised MDP (Wang et al., 2022) in enhancing task-centric representations. However, the two approaches are entirely irrelevant. In Fig 22, we show a comparison of our method to Denoised MDP. Denoised MDP performs information decomposition by changing the MDP transition structure and utilizing the reward as a guide to separate task-relevant information. However, since Denoised MDP does not modify the weight for the reward modeling task, the observation modeling task can still dominate the learning process. Consequently, the training signals from the reward modeling task may be inadequate to guide decomposition. It's also worth

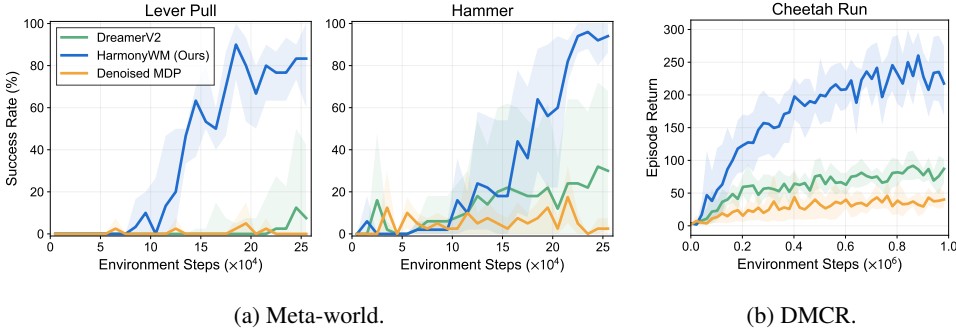

(a) Meta-world.

(b) DMCR.

Figure 22: Comparison of HarmonyWM with Denoised MDP.

noting that Denoised MDP only added noise distractors to task-irrelevant factors in their DMC experiments. On the other hand, the benchmark adopted in our experiments, DMCR, adds visual distractors to both task-irrelevant and task-relevant factors, such as the color of the body and floor, which adds complexity to both factors and results in more challenging tasks. These two reasons above can account for the low performance of Denoised MDP in our benchmarks.

## D.10 ATARI100K EXPERIMENTS

In Fig 23, we present our result of Harmony DreamerV3 on the Atari100K benchmark. The visual observation complexity in Atari games is less demanding compared to visual manipulation and locomotion domains. As a result, the issue of observation modeling domination is not as pronounced. However, our method can still enhance the base DreamerV3 performance on more intricate tasks, such as Qbert.

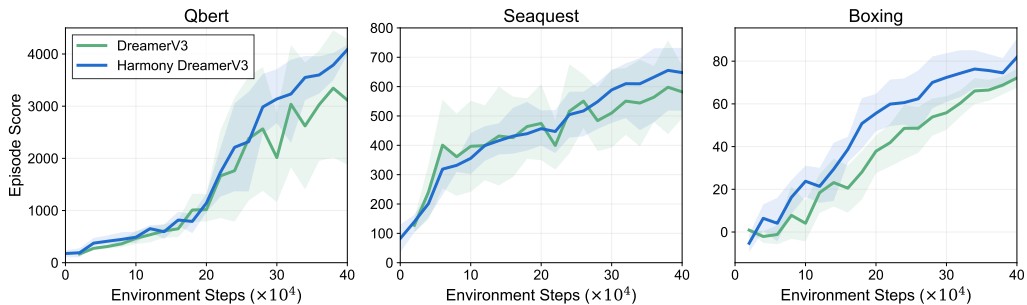

Figure 23: Performance of Harmony DreamerV3 on Atari100K.

## D.11 QUANTITATIVE EVALUATION OF THE BENEFICIAL IMPACT OF OBSERVATION MODELING ON REWARD MODELING

To explore the possible beneficial impact of observation modeling on reward modeling, we utilize the offline experimental setup in Fig 4 and 5, whose details are described in Appendix C.4. We offline train two DreamerV2 agents using task weights $(w_r = 100, w_o = 1)$ and $(w_r = 100, w_o = 0)$ and evaluate the ability to accurately predict rewards on a validation set with the same distribution as the offline training set. For this task, we gathered 20,000 segments of trajectories, each of length 50. We utilized 35 frames for observation and predicted the reward for the remaining 15 frames. Results are reported in the form of average MSE loss. We observe that the world model with observation modeling predicts the reward better than the world model that only models the reward. The prediction loss of $(w_r = 100, w_o = 1)$ is 0.379, while the loss of $(w_r = 100, w_o = 0)$ is 0.416. This result indicates that observation modeling has a positive effect on reward modeling.

