# OpenReview forum: "Harmony World Models: Boosting Sample Efficiency for Model-based Reinforcement Learning"
_ICLR.cc/2024/Conference — Submitted to ICLR 2024_

### Official Review · Reviewer_pH7N · 2023-10-26

**Soundness:** 2 fair
**Presentation:** 3 good
**Contribution:** 2 fair
**Rating:** 5
**Confidence:** 4

**Summary:**

Harmony World Models: Boosting Sample Efficiency for Model-based Reinforcement Learning

In this paper, the authors observe that the coefficients of reward loss and dynamic loss, if not chosen carefully, could affect the performance of the final model-based RL algorithm.
And the authors propose to use a set of learnable parameters to approximate loss scale to balance the loss.
There are performance improvements introduced as shown in the experiments.

**Strengths:**

1. The paper is well written and easy to understand.
2. The proposed method is quite straight-forward, making it reproducible and easy to apply to existing algorithms.

**Weaknesses:**

1. There’s no good ablation experiments on different reward blending schemes.

    The chosen method for blending/weighting the loss terms looks reasonable, but not necessarily the best method or even the most reliable one.

    There needs to be some other weighting methods tested in the experiment sections.

    Some simple weighting methods should be compared, which include for example the one in [1] and some other straight-forward heuristic methods.

2. It would also be helpful to show how sensitive the proposed method is.

    Does it require separate tuning for each task?

    If it needs to be tuned, how much effort is needed? Is it stable across random seeds?

3. More baselines and environments are needed.

    The proposed method is only compared against dreamerV2 in a small subset of tasks.

    What will happen if it is compared to MuZero, TD-MPC and SimPle etc as mentioned in the paper?

    And what will happen if the algorithm is tested in similar environments such as GO, those atari environments or locomotion control tasks from image?

**Questions:**

The performance of DreamerV2 seems too bad in some environments like Cheetah Run, Walker Run, which is hard to figure out why considering DreamerV2 was applied to far harder problems and succeeded and DreamerV1 has better performance with the same number of training samples.

Why does it happen?

[1] Kendall, Alex, Yarin Gal, and Roberto Cipolla. "Multi-task learning using uncertainty to weigh losses for scene geometry and semantics." In Proceedings of the IEEE conference on computer vision and pattern recognition, pp. 7482-7491. 2018.

---

> ### Author Response · Authors · 2023-11-16
> **Response to Reviewer pH7N**
>
> We sincerely thank Reviewer pH7N for providing a detailed review and insightful questions.
>
> **Q1**: Comparisons to different reward blending schemes
>
> We want to clarify that **the major contribution of our paper is to reveal the overlooked potential of sample efficiency in MBRL through a multi-task perspective, not to propose a specific multi-task method**. Our experiments find that our straightforward method is dramatically effective and sufficient to address the problem revealed in our paper.
>
> We have compared with different multi-task methods and discussed why these methods can hardly make more improvements in the problem of our paper. Please refer to $\underline{\text{Q2 of the global response}}$ for the detailed response.
>
> **Q2**: Sensitivity of HarmonyWM
>
> **Our HarmonyWM does not add hyperparameters** to our base method and, therefore, does not need additional tuning for different tasks. The improvement is stable across random seeds, as shown in the confidence intervals reported in our figures.
>
> **Q3**: Additional baselines and environments
>
> Inevitably, the scale of our experiments is restricted due to our limited computational resources. We have conducted experiments on 11 (Metaworld, RLBench, DMCR) and 4 environments (DMC) in the main paper and appendix of the original submission. During rebuttal, we added 4 additional environments (Assembly in Metaworld and Qbert/Seaquest/Boxing in Atari). The number of experimental environments in our work has been on par with or larger than most recent publications on MBRL (please refer to $\underline{\text{response to Q3 of Reviewer 72ZS}}$).
>
> Regarding compared baselines, since our HarmonyWM is based on DreamerV2, it is natural to compare it mainly with DreamerV2 to reveal the significance of harmonizing loss scales among world model learning tasks. We mention MuZero, TD-MPC and SimPLe in Figure 1 to show the high-level spectrum of world model learning. Some model-based methods are tailored to a specific area and therefore unfair to conduct head-to-head comparisons.
>
> Regarding the environments Reviewer pH7N proposed, it is beyond our computational resources to conduct experiments on GO, and also unfair since MuZero with MCTS is mainly tailored to board games and unable to handle continuous control, while both Dreamer and HarmonyWM are more general architectures. For the Atari domain, SimPLe is a weak baseline and is outperformed by DreamerV3, therefore we perform additional comparisons to DreamerV3. The corresponding results can be found in $\underline{\text{Fig. 23 of Appendix D.10}}$ of our latest revision. For locomotion control tasks from images, we have already presented comparisons to TD-MPC and DreamerV2 on the DMCR domain. Results can be found in $\underline{\text{Fig. 10(b) of main paper}}$ and $\underline{\text{Fig. 18 of Appendix D.5}}$.
>
>
> **Q4**: DreamerV2 performance on DMC Remastered benchmark
>
> Regarding the comment that DreamerV2 performs poorly in environments like Cheetah Run and Walker Run, we need to clarify that we use the **DMC Remastered** benchmark in our main paper, **which is different from and more challenging than the original DMC**, as it contains various random visual distractors, showcased in the $\underline{\text{left of Fig 8 of main paper}}$, and has proven difficult for model-based agents to learn on. DreamerV2 performs well on the original DMC benchmark, as shown in $\underline{\text{Fig. 17 of Appendix D.5}}$, but performs poorly on DMC Remastered as it suffers from the domination of reconstructing irrelevant visual distractors in world model learning.
>
> Please let us know whether our response addresses all your concerns. We are more than happy to discuss more if you have further questions or more suggestions to improve our paper.

---

> ### Author Response · Authors · 2023-11-21
> **Request of Reviewer's attention and feedback**
>
> Dear Reviewer pH7N,
>
> Thanks again for your dedication to reviewing our paper.
>
> We write to kindly remind you that this is the last few days of the reviewer-author discussion period. We have made every effort to address the concerns you suggested and improve our paper:
>
> * We have made comprehensive comparisons to other weighting methods and discussed why these methods can hardly make more improvements in the problem of our paper.
> * We clarify that our method does not add hyperparameters, and the improvement is stable across random seeds.
> * We perform additional experiments on the Atari domain proposed by you and show that even though the issue of observation modeling domination is not as pronounced in the Atari domain, our method can still enhance the base method's performance on more intricate tasks.
> * We clarify that we use the DMC Remastered benchmark in our main paper, which is different from and more challenging than the original DMC. This factor accounts for the poor performance of DreamerV2 on the DMCR Cheetah Run and Walker Run tasks.
>
> Please kindly let us know if you have any remaining questions. If our responses have addressed your concerns, would you please consider re-evaluating our work based on the updated information? Looking forward to your reply.
>
> Sincerely,
>
> Authors

---

> ### Author Response · Authors · 2023-11-22
> **Discussion period ends soon**
>
> Dear Reviewer pH7N,
>
> Thank you once more for your invaluable review.
>
> It is a kind reminder that **this is the last day of the 12-day Reviewer-author discussion period**. Taking your insightful suggestions seriously, we have made every effort to provide all the experiments and clarifications that we can.
>
> If you have read our response, **we would greatly appreciate your acknowledgment and re-evaluation of our work**. We remain eagerly available for any further questions or discussions you may have. Anticipating your reply.
>
> Best regards,
>
> Authors.

---

### Official Review · Reviewer_WtCd · 2023-10-30

**Soundness:** 3 good
**Presentation:** 3 good
**Contribution:** 2 fair
**Rating:** 6
**Confidence:** 4

**Summary:**

This paper identifies the multi-task essence (the dynamics and the reward) of world models and analyzes the disharmonious interference between different tasks. The authors empirically find that adjusting the coefficient of reward loss (which is overlooked by most previous works) can emphasize task-relevant information and improve sample efficiency. To adaptively balance the reward and the dynamic losses, the authors propose a harmonious loss function that learns a global reciprocal of the loss scale using uncertainty. The experiments show that the proposed method outperforms Dreamerv2 with a considerable gap and has better generality than DreamerV3 and DreamerPro.

**Strengths:**

This paper systematically identifies the multi-tasking essence of world models, which is essential to model-based methods and is overlooked by most previous work. The whole paper is well-written and easy to follow.

> The proposed method has generality to other model-based RL algorithms and significantly improves the sample efficiency.

**Weaknesses:**

> The novelty of the proposed method is limited. The authors build the optimization process of world models as a multi-task optimization and then use uncertainty weighting to harmonize loss scales among tasks. More state-of-the-art methods, e.g., multi-objective optimization or multi-agent learning methods, can be considered and may further improve the performance.

**Questions:**

> In Fig.15 & 16, HarmonyWM $ w_d=1 $ learns faster than Harmony WM at the beginning of some tasks. Could the authors give some explanations about this?

> The authors claim "the root cause as the disharmonious interference between two tasks in explicit world model learning: due to *overload of redundant observation signals* ...". Why does, for example, Denoised MDP, which identifies the information that can be safely discarded as noises, serve as a baseline in the experiments?

>  Is there a sufficient reason why the reward modeling and the observation modeling tasks are scaled to the same constant? Should we maintain the equilibrium between these tasks? Or can we emphasize some tasks at some specific states?

---

> ### Author Response · Authors · 2023-11-16
> **Response to Reviewer WtCd**
>
> We sincerely thank Reviewer WtCd for providing a detailed review, insightful questions, and a positive evaluation of our paper.
>
> **Q1**: Limited novelty of the proposed method
>
> We want to clarify that **the major contribution of our paper is to reveal the overlooked potential of sample efficiency in MBRL through a multi-task perspective**, and to propose a decent way to address this. We acknowledge that our method for balancing different tasks in world models is straightforward, but as pointed out by Reviewer pH7N, **this simplicity serves as an advantage, making it easy to reproduce and apply to existing algorithms**.
>
> Regarding the comment that we use uncertainty weighting to harmonize loss scales, we would like to note that our method differs from uncertainty weighting in several aspects. Uncertainty weighting is derived from maximum likelihood estimation, while our motivation is to harmonize loss scales among tasks. Uncertainty weighting makes assumptions on the distributions behind losses, while our method removes these assumptions and makes it possible to also balance the KL loss. Moreover, we introduce a rectified version to avoid extremely large task weights. Please refer to the $\underline{\text{last paragraph of Sec 3 of main paper in original submission}}$ for a more detailed explanation on this.
>
>
> **Q2**: Comparison to multi-task or multi-objective baselines
>
> Our work is irrelevant to multi-agent learning methods, as the multi-task perspective is presented for understanding the world model learning part of MBRL, instead of the policy learning part.
>
> Regarding multi-task learning methods, we have conducted comparison experiments and detailed discussions. Please refer to $\underline{\text{Q2 of the global response}}$.
>
> **Q3**: Comparison to Denoised MDP
>
> Although we share a similar point with Denoised MDP to enhance task-centric representations, our approach is entirely irrelevant to it and has the advantage of more simplicity. Our contributions regarding world model optimization are **orthogonal** to Denoised MDP regarding architecture and can be combined with it to further improve performance, which is left for future work.
>
> Nevertheless, we present additional comparisons to Denoised MDP in $\underline{\text{Fig. 22 of Appendix D.9 of latest revision}}$, where Denoised MDP trails behind our HarmonyWM.  Denoised MDP performs information decomposition by changing the MDP transition structure and utilizing the reward as a guide to separate controllable information. However, since Denoised MDP does not modify the weight for the reward modeling task, the observation modeling task can still dominate the learning process. Consequently, the training signals from the reward modeling task may be inadequate to guide decomposition.
>
> **Q4**: Ablation on adjusting $w_d$
>
> We observe that HarmonyWM ($w_d=1$) learns faster than HarmonyWM at the beginning of the Hammer task in terms of success rate. We argue that this is because in this task, high success rates do not strictly correspond to high return. The hammer task can be hacked by pushing the nail using the arm instead of the hammer, which results in the task's success with a low reward. When measured in terms of episode returns, HarmonyWM performs comparably with HarmonyWM ($w_d=1$) at the beginning and slightly outperforms HarmonyWM ($w_d=1$) at the end of training.
>
> **Q5**: Necessity of scaling different tasks to the same constant
>
> As we demonstrated in our analysis ($\underline{\text{Sec 2.3 of main paper}}$), domination of either task in world model learning cannot fully exploit the potential of sample efficiency; thus, we do need to make a proper balance these two tasks in world model learning. We find it is dramatically effective to simply maintain the equilibrium between them, i.e., rescaling losses to the same constant. It is interesting to explore a more fine-grained balancing strategy, for example, as suggested by the reviewer, to emphasize some tasks a bit more at some specific states. This is left for future work.
>
> Please let us know whether our response addresses all your concerns. We are happy to discuss more if you have further questions or suggestions to improve our paper.

---

> ### Author Response · Authors · 2023-11-21
> **Request of Reviewer's attention and feedback**
>
> Dear Reviewer WtCd,
>
> Thanks again for your dedication to reviewing our paper.
>
> We write to kindly remind you that this is the last few days of the reviewer-author discussion period. We have made every effort to address the concerns you suggested and improve our paper:
>
> * We clarify that a major contribution of our paper is to reveal the overlooked potential of sample efficiency in MBRL through a unified multi-task perspective into world model learning, besides proposing a decent way to address this. We also note that our method differs from uncertainty weighting in several aspects.
> * We have made comprehensive comparisons to multi-task baselines and discussed why these methods can hardly make more improvements in the problem of our paper.
> * We clarify that Denoised MDP is orthogonal to our technical contributions. Nevertheless, we provide additional comparison results against Denoised MDP and explain the possible reasons for its inferior performance.
> * We replied to the questions you mentioned in the Questions part.
>
> Please kindly let us know if you have any remaining questions. If our responses have addressed your concerns, would you please consider re-evaluating our work based on the updated information? Looking forward to your reply.
>
> Sincerely,
>
> Authors

---

> ### Author Response · Authors · 2023-11-22
> **Discussion period ends soon**
>
> Dear Reviewer WtCd,
>
> Thank you once more for your invaluable review.
>
> It is a kind reminder that **this is the last day of the 12-day Reviewer-author discussion period**. Taking your insightful suggestions seriously, we have made every effort to provide all the experiments and clarifications that we can.
>
> If you have read our response, **we would greatly appreciate your acknowledgment and re-evaluation of our work**. We remain eagerly available for any further questions or discussions you may have. Anticipating your reply.
>
> Best regards,
>
> Authors.

---

### Official Review · Reviewer_72ZS · 2023-10-30

**Soundness:** 3 good
**Presentation:** 3 good
**Contribution:** 2 fair
**Rating:** 3
**Confidence:** 3

**Summary:**

This paper proposes viewing model learning from a multi-task perspective and introduces a method to adjust the trade-off between observation loss and reward loss to train the world model. Combined with DreamerV2, the proposed Harmony World Model achieves a noticeable performance improvement over DreamerV2 in several domains.

**Strengths:**

1. The proposed method is novel and interesting.
2. Compared to DreamerV2, there is a significant improvement in performance.
3. The paper is well-written and easy to follow.

**Weaknesses:**

1. Although the method proposed in this paper is novel, DreamerV3 has already addressed the issue of differing scales in reconstructing inputs and predicting rewards using Symlog Predictions. This paper primarily conducts extensive experiments in comparison with DreamerV2 and lacks more comparisons and discussions with DreamerV3. This significantly diminishes the contributions of this paper.

2. Moreover, as TDMPC is mentioned in the paper as one of the Implicit MBRL methods, I believe it should also be considered as one of the important baselines for comparisons across different benchmarks.

3. The experimental environments chosen in the paper are all easy tasks from different domains. I think experiments should be conducted in more challenging environments, such as Hopper-hop and Quadruped-run in DMC, Assembly and Stick-pull in MetaWorld, etc. Moreover, given the current trend in the community to conduct large-scale experiments, having results from only eight environments seems somewhat limited.

**Questions:**

See weaknesses above.

---

> ### Author Response · Authors · 2023-11-16
> **Response to Reviewer 72ZS (Part 1)**
>
> Many thanks to Reviewer 72ZS for providing a thorough review and valuable comments.
>
> **Q1**: Clarifications and comparison with DreamerV3
>
>
> Reviewer 72ZS commented that DreamerV3 has already *addressed the issue of differing scales in reconstructing inputs and predicting rewards using Symlog Predictions*. We need to clarify that **using Symlog Predictions does not solve our problem of seeking a balance between reward modeling and observation modeling**. First of all, the Symlog transformation only shrinks extremely large values but is unable to rescale various values into exactly the same magnitude, while our harmonious loss properly addresses this by dynamically approximating the reciprocals of the values. More importantly, the primary reason why $L_r$ has a significantly smaller loss scale is the difference in dimension: as we have stated in our main paper, the observation loss $L_o$ usually aggregates $H\times W\times  C$ dimensions, while the reward loss $L_r$ is derived from only a scalar (a detailed explanation of why $L_o$ is summed over dimensions can be found in $\underline{\text{response to Q3.4 of Reviewer QvDA}}$ ). In summary, **using Symlog Predictions as DreamerV3 only mitigates the problem of differing per-dimension scales** (typically across environment domains) by a static transformation, **while our method aims to dynamically balance the overall loss scales across tasks in world model learning, considering together per-dimension scales, dimensions, and training dynamics**.
>
> In practice, DreamerV3 does not use Symlog predictions in reconstructing visual inputs due to unified scales of visual inputs across domains. Besides, DreamerV3 uses twohot symlog predictions for the reward predictor to replace the MSE loss in DreamerV2. This approach increases the scale of the reward loss, but is insufficient to mitigate the domination of the observation loss. We observe that the reward loss in DreamerV3 is still significantly smaller than the observation loss, especially for visually demanding domains such as RLBench, where the reward loss is still two orders of magnitude smaller.
>
> Overall, **our contributions are orthogonal to the modifications in DreamerV3** and can be combined with DreamerV3 (which we brand as Harmony DreamerV3) to improve performance further. We have already presented the results of our Harmony DreamerV3 in $\underline{\text{Fig. 9 of main paper in the original submission}}$, and we provide additional results in $\underline{\text{Fig. 20 of Appendix D.7}}$ of our latest revision. We apologize for the limited discussion on DreamerV3, and we have added a discussion section in $\underline{\text{Appendix D.7 of our latest revision}}$.
>
> **Q2**: Comparison with TD-MPC
>
> We want to highlight that we have already **presented aggregated results of TD-MPC on three Meta-world tasks** in $\underline{\text{Fig. 10(b) of main paper}}$. Moreover, we present **the individual learning curves of TD-MPC compared to our HarmonyWM and DreamerV2 on six Meta-world and DMCR tasks** in $\underline{\text{Fig. 18 of Appendix D.5 in the original submission}}$. These results show that TD-MPC is unable to learn a meaningful policy in some tasks, which further highlights the value of our explicit-implicit method.

---

> ### Author Response · Authors · 2023-11-16
> **Response to Reviewer 72ZS (Part 2)**
>
> **Q3**: Difficulties and scales of experimental environments
>
> We **respectfully disagree with the comment that our experimental environments are all easy tasks**. According to the classification standard of Seo et al. [1], ***Sweep-Into*, *Hammer* and *Push* can all be regarded as challenging environments** in the Meta-world domain. The RLBench domain itself is considered more demanding than Meta-world due to its realistic visual observations. It's also worth noting that we use **DMC Remastered, which is different from and more challenging than the original DMC**, as it contains various random visual distractors and has proven difficult for model-based agents to learn on [6]. More details about the environments we use can be found in $\underline{\text{Appendix C.1}}$, and we apologize for not referring to these in our main paper.
>
> Regarding the challenging environments proposed by reviewer 72ZS, we have already reported our results on DMC Quadruped-run in $\underline{\text{Fig. 17 of Appendix D.5}}$, along with several other DMC environments. We apologize for not referring to them in our main paper. **The *Push* task reported in our main paper is of similar difficulty to *Assembly* and *Stick-pull*.** Nevertheless, we provide additional results on *Assembly* in $\underline{\text{Fig. 19 of Appendix D.6}}$ of our latest revision.
>
> Regarding the comment that our experiments lack variety and that we should perform large-scale experiments, we need to clarify that we perform **our experiments on fifteen instead of eight environments**: six from Meta-world, two from RLBench, three from DMC Remastered, and four from standard DMC (in Appendix). We add another 4 environments (Assembly in Metaworld and Qbert/Seaquest/Boxing in Atari) in our latest revision, which sums to a total of 19. Inevitably, the scale of our experiments is restricted due to our limited computational resources. However, this scale is very common in recent model-based RL publications, such as APV (9 envs) [2], DreamerPro (12 envs) [3], IsoDream (5 envs) [4], Curious Replay (5 envs) [5], IPV (12 envs) [6] and so forth.
>
> [1] Seo et al. Masked world models for visual control. CoRL, 2022.
>
> [2] Seo et al. Reinforcement learning with action-free pre-training from videos. ICML, 2022.
>
> [3] Deng et al. Dreamerpro: Reconstruction-free model-based reinforcement learning with prototypical representations. ICML, 2022.
>
> [4] Pan et al. Iso-Dream: Isolating and Leveraging Noncontrollable Visual Dynamics in World Models. NeurIPS, 2022.
>
> [5] Kauvar et al. Curious Replay for Model-based Adaptation. ICML, 2023.
>
> [6] Wu et al. Pre-training contextualized world models with in-the-wild videos for reinforcement learning. NeurIPS, 2023.
>
> Please let us know whether our response addresses all your concerns. We are happy to discuss more if you have further questions or suggestions to improve our paper.

---

> ### Author Response · Authors · 2023-11-21
> **Request of Reviewer's attention and feedback**
>
> Dear Reviewer 72ZS,
>
> Thanks again for your dedication to reviewing our paper.
>
> We write to kindly remind you that this is the last few days of the reviewer-author discussion period. We have made every effort to address the concerns you suggested and improve our paper:
>
> * We clarify that using Symlog Predictions in DreamerV3 does not address our problem and that the modifications of DreamerV3 are orthogonal to our technical contributions. Nevertheless, we provide additional comparison results against DreamerV3, where our method still proves effective.
> * We clarify that we have already presented the results of TD-MPC in our original submission, and results show that TD-MPC cannot learn a meaningful policy in some tasks, further highlighting the value of our explicit-implicit method.
> * We explain in detail the environments we use in our paper and that they are all challenging tasks. We also provide additional results on challenging environments proposed in your review. The results further demonstrate the superiority of our method on difficult tasks.
> * We clarify the number of environments we use and compare it to that of recent model-based RL publications to show that the scale of our experiments is sufficient.
>
> Please kindly let us know if you have any remaining questions. If our responses have addressed your concerns, would you please consider re-evaluating our work based on the updated information? Looking forward to your reply.
>
> Sincerely,
>
> Authors

---

> ### Author Response · Authors · 2023-11-22
> **Discussion period ends soon**
>
> Dear Reviewer 72ZS,
>
> Thank you once more for your invaluable review.
>
> It is a kind reminder that **this is the last day of the 12-day Reviewer-author discussion period**. Taking your insightful suggestions seriously, we have made every effort to provide all the experiments and clarifications that we can.
>
> If you have read our response, **we would greatly appreciate your acknowledgment and re-evaluation of our work**. We remain eagerly available for any further questions or discussions you may have. Anticipating your reply.
>
> Best regards,
>
> Authors.

---

### Official Review · Reviewer_QvDA · 2023-10-31

**Soundness:** 3 good
**Presentation:** 3 good
**Contribution:** 2 fair
**Rating:** 6
**Confidence:** 3

**Summary:**

Typical MBRL methods optimize for two tasks - observation modeling (aka explicit MRL) and reward modeling (aka implicit MBRL). The paper hypothesizes that there is interference between the two tasks and proposed an alternate scheme for weighing the losses corresponding to the two tasks.

**Strengths:**

1. The paper is well written and flows nicely.

2. The authors experiment with 4 set of environments and the results look reasonable

**Weaknesses:**

1. The paper calls out the following as it contribution: "To the best of our knowledge, our work, for the first time, systematically identifies the multitask essence of world models and analyzes the disharmonious interference between different tasks, which is unexpectedly overlooked by most previous work". While I agree that MBRL can be formulated could be multi-task RL problem (it is clearly multi-objective problem), the paper does not do a through job at analyzing "the disharmonious interference between different tasks". e.g. they do not study if and why there is an interference between different tasks. Note that in the multi-task literature, interference often refers to progress on one task, hindering the progress of another task. What the authors show is that adhoc setting of scalars for the different losses can hurt the performance on the RL task but they do not show that it hurts the performance on the tasks being directly optimized for. The distinction is important in the multi-task setup (which is what the paper uses).

2. While the paper formulates the problem as a multi-task problem, they do not compare with any multi-task baseline that could balance between the different losses in equation 3. So while they show that adjusting the loss coefficients help, they do not show if their approach is better than other multi-task approaches.

**Questions:**

Listing some questions (to make sure I better understand the paper) and potential areas of improvement. Looking forward to engaging with the authors on these questions and the points in the weakness section.

1. In equation 1, the input to the representation model $q_{\theta}$ should be $o_t$ right ?
2. The paper seems to suggest that "observation modeling" task is a new task. Is that correct ?
3. The word "harmony" in HWM - does it come from some mathematical properties or it refer to "harmony" (tuning) between the losses?
4. Regarding "the scale of L r is two orders of magnitude smaller than that of L o, which usually aggregates H × W × C dimensions", it usually doesnt matter what the dimensions of output are as the loss is averaged over the dimensions.
5. Are the findings 1, 2, 3 are new findings ?

---

> ### Author Response · Authors · 2023-11-16
> **Response to Reviewer QvDA**
>
> Many thanks to Reviewer QvDA for providing a thorough review and valuable questions.
>
> **Q1**: Clarification on the multi-task essence of world models
>
> We apologize for using the misleading word 'interference' to describe the problem in our original submission. In our work, we conduct a dedicated investigation ($\underline{\text{Sec 2.3 of main paper}}$) and demonstrate that observation modeling and reward modeling can benefit from each other, as shown in $\underline{\text{Fig 2 of main paper}}$ and our additional quantitative results in $\underline{\text{Q1 of the global response}}$. More importantly, we also reveal that domination of either task, as in many existing MBRL literature, does not properly exploit these multi-task benefits. Thus, the **main problem our work addresses can be more appropriately named task 'domination'** instead of 'interference,' and we propose a simple yet effective strategy to dynamically balance tasks in world models. We have modified our wording in the text of our $\underline{\text{latest revision}}$, and thank you for pointing out this.
>
> Please also refer to $\underline{\text{Q1 of the global response}}$ for the detailed response.
>
> **Q2**: Comparison to multi-task baselines
>
> We have compared different multi-task methods and discussed why these methods can hardly make more improvements in the problem of our paper. Please refer to $\underline{\text{Q2 of the global response}}$ for the detailed response.
>
> **Q3**: Clarifications and minor questions
>
> 1. The input to the representation model $q_\theta$ is the previous latent $z_{t-1}$, the action $a_{t-1}$, and current observation $o_t$, as formulated in our main paper. Detailed architecture of Dreamer has been illustrated in $\underline{\text{Fig 6 of main paper}}$, while Fig 1 only presents a simplied one. You may refer to 4 below or Dreamer's original papers [1,2] for additional information on this.
> 2. Observation modeling is not a new task. As shown in $\underline{\text{Fig 1 of main paper}}$, while it is not necessary for MBRL, the observation modeling task has always been a dominating task in explicit MBRL methods facilitating representation learning. Our work emphasizes that previous literature has overlooked the multi-tasking essence of world model learning and that proper mitigation of task dominations can significantly improve MBRL.
> 3. The word 'harmony' here refers to harmony between the losses (tasks), where none of the two tasks dominates in world model learning, and either task can benefit from the other's progress.
> 4. The observation loss $L_o$ is computed as a sum over all dimensions, instead of an average. The world model in the Dreamer framework is formulated as sequential variational inference [1] and optimized by the standard variational bound on log likelihood (ELBO): $
> \mathcal{L}(\theta) =\mathbb{E}\_{q\_{\theta}\left(z_{1:T}|a_{1:T}, o_{1:T}\right)} \Big[ \sum\_{t=1}^{T} \Big(
>     {-\log p\_{\theta}(o\_{t}|z\_{t}) -\log p\_{\theta}(r\_{t}|z\_{t}) } {+\text{KL}\left[q\_{\theta}(z\_{t}|z\_{t-1},a\_{t-1}, o\_{t}) \Vert p\_{\theta}(\hat{z}\_{t}|z\_{t-1}, a\_{t-1}) \right]}
>     \Big)\Big],$ where the observation loss $L_o=-\log p_{\theta}(o_{t}|z_{t})=-\sum_{i=1}^{h}\sum_{j=1}^w \log p_{\theta}(o_{t}^{(i,j)}|z_{t})$ is natually in the form of a sum over pixels $o_{t}^{(i,j)}$. Averaging over dimensions is equivalent to assigning a very small weight to the observation loss and leads to KL term domination and inferior results.
> 5. To the best of our knowledge, findings 1, 2, 3 are new for world model learning. We have already discussed the differences between our findings and previous relevant literature in the $\underline{\text{last paragraph of Sec 2 in the original submission}}$.
>
> [1] Hafner et al. Learning latent dynamics for planning from pixels. ICML 2019.
>
> [2] Hafner et al. Dream to control: Learning behaviors by latent imagination. ICLR 2020.
>
> Please let us know whether our response addresses all your concerns. We are happy to discuss more if you have further questions or suggestions to improve our paper.

---

> ### Author Response · Authors · 2023-11-21
> **Request of Reviewer's attention and feedback**
>
> Dear Reviewer QvDA,
>
> Thanks again for your dedication to reviewing our paper.
>
> We write to kindly remind you that this is the last few days of the reviewer-author discussion period. We have made every effort to address the concerns you suggested and improve our paper:
>
> * We clarify our multi-task perspective into world models and stress that a major contribution of our paper is to reveal the overlooked potential of sample efficiency in MBRL through a unified multi-task perspective into world model learning.
> * We clarify our misleading word 'interference' and have modified our wording to 'domination' in our latest revision.
> * We have made comprehensive comparisons to multi-task baselines and discussed why these methods can hardly make more improvements in the problem of our paper.
> * We replied to the questions you mentioned in the Questions part.
>
> Please kindly let us know if you have any remaining questions. If our responses have addressed your concerns, would you please consider re-evaluating our work based on the updated information? Looking forward to your reply.
>
> Sincerely,
>
> Authors

---

> ### Author Response · Authors · 2023-11-22
> **Discussion period ends soon**
>
> Dear Reviewer QvDA,
>
> Thank you once more for your invaluable review.
>
> It is a kind reminder that **this is the last day of the 12-day Reviewer-author discussion period**. Taking your insightful suggestions seriously, we have made every effort to provide all the experiments and clarifications that we can.
>
> If you have read our response, **we would greatly appreciate your acknowledgment and re-evaluation of our work**. We remain eagerly available for any further questions or discussions you may have. Anticipating your reply.
>
> Best regards,
>
> Authors.

---

### Author Response · Authors · 2023-11-16
**Global Response to All Reviewers**

We appreciate the thorough comments and insightful feedback from the reviewers. We have made every effort to address all the reviewers' concerns and responded to the individual reviews below. We have also answered the common questions raised by the reviewers in this global response.

**Q1**: Clarification of our multi-task perspective into world models

One major contribution of our paper is to **reveal the overlooked potential of sample efficiency in MBRL through a unified multi-task perspective into world model learning**. Two inherent tasks are identified in world model learning: observation modeling and reward modeling. The former dominates in explicit MBRL methods, while the latter dominates in implicit MBRL. To our knowledge, no previous literature explicitly pointed out this.

Typically, two kinds of task relationships exist in multi-task learning: collaboration or conflict. In the collaboration case, one task can benefit from joint learning with the others, while in the conflict case, learning one task can hurt the performance of the others. We find that **there is more collaboration in world model learning**: the reward modeling task can encourage the observation modeling task to focus on task-relevant details and thus make more accurate predictions, while the observation modeling task simultaneously serves an auxiliary task to stabilize and accelerate the reward modeling task due to relatively sparse reward signals. Our qualitative evaluation in $\underline{\text{Fig 2 of main paper}}$ and additional quantitative evaluation below support our above claims.

However, we reveal that standard practice in many MBRL methods does not properly leverage this multi-task essence of world models: domination of either observation modeling or task modeling can not fully unleash the aforementioned positive benefits among these two tasks. Thus, another contribution of our work is to **introduce a simple yet effective strategy to dynamically balance these two tasks, which can be easily integrated to accelerate a line of MBRL methods**. While our method is straightforward, our insights on the problem and the generality of our approach are valuable.

**Quantitative evaluation**: To explore the possible beneficial impact of observation modeling on reward modeling, we offline train two DreamerV2 agents using task weights $(w_r=100, w_o=1)$ and $(w_r=100, w_o=0)$ and evaluate the ability to accurately predict rewards. We observe that the world model with observation modeling predicts the reward ~10% better than the world model that only models the reward, which indicates that observation modeling has a positive effect on reward modeling. Please refer to $\underline{\text{Appendix D.11 of latest revision}}$ for details on this experiment.

---

> ### Author Response · Authors · 2023-11-16
> **Global Response to All Reviewers (Cont'd)**
>
> **Q2**: Discussion and comparison to multi-task methods
>
> We acknowledge that there are many advanced methods in the field of multi-task learning or multi-objective learning. They can be roughly categorized into loss-based and gradient-based methods. Since gradient-based methods mainly address the problem of gradient conflicts [1,2], which is not the main case in world model learning, we focus on loss-based methods, which assign different weights to task losses by various criteria. We choose the following as our baselines to discuss differences and conduct comparison experiments:
>
> 1. Uncertainty weighting (UW) [3] balances tasks with different scales of targets, which is measured as uncertainty of outputs. As pointed out in our paper, in world model learning, observation loss $\mathcal{L}\_o(\theta)=-\log p\_\theta\left(o\_t \mid z\_t\right)=-\sum\_{i=1}^{h}\sum\_{j=1}^w \log p\_{\theta}(o\_{t}^{(i,j)}|z\_{t})$ and reward loss $\mathcal{L}\_r(\theta)=-\log p\_\theta\left(r\_t \mid z\_t\right)$ differs not only in scales but also in dimensions. Please refer to $\underline{\text{last paragraph of Sec 3 in main paper}}$ for more discussions on this.
> 2. Dynamics weight average (DWA) [4] balances tasks according to their learning progress, illustrating the various task difficulties. However, in world model learning, since the data in the replay buffer is growing and non-stationary, the relative descending rate of losses may not accurately measure task difficulties and learning progress.
> 3. NashMTL [5] is the most similar to our method, whose optimization direction has balanced projections of individual gradient directions. However, its implementation is far more complex than ours, as it introduces an optimization procedure to determine loss weights on each iteration. In our experiments, we also find this optimization is prone to deteriorate to produce near-zero weights without careful tuning tuning of optimization parameters. Moreover, compared to HarmonyWM introducing a rectified loss, NashMTL does nothing to avoid extreme task weighting, which we actually encounter in our experiments.
>
> **Experimental results**: As shown in $\underline{\text{Fig 21 in Appendix D.8 of latest revision}}$, our method is the most effective one among multi-task methods, which also has the advantage of simplicity.
>
> [1] Yu et al. Gradient surgery for multi-task learning. NeurIPS 2020.
>
> [2] Liu et al. Conﬂict-Averse Gradient Descent for Multi-task Learning. NeurIPS 2021.
>
> [3] Kendall et al. Multi-Task Learning Using Uncertainty to Weigh Losses for Scene Geometry and Semantics. CVPR 2018.
>
> [4] Liu et al. End-to-end multi-task learning with attention. CVPR 2019.
>
> [5] Navon et al. Multi-task learning as a bargaining game. ICML 2022.

---

### Author Response · Authors · 2023-11-16
**Summary of Revision**

We would like to thank the reviewers for their detailed comments. In addition to a global response and individual responses to each review, we've also updated the paper with a number of modifications to address reviewer suggestions and concerns. Summary of updates:

1. We clarified our multi-task perspective of world models in the Introduction section, and unified the problem as 'domination of a particular task' in world model learning throughout the paper, in response to **Reviewer QvDA's** feedback.
2. We added additional results of HarmonyWM on one challenging task, Assembly from Meta-world (**Appendix D.6**), fulfilling a request from **Reviewer 72ZS**.
3. We added additional results of Harmony DreamerV3 on two tasks from Meta-world and expanded the discussion to clarify that our contribution is orthogonal to DreamerV3 (**Appendix D.7**), in response to **Reviewer 72ZS's** feedback.
4. We added comparisons to multi-task learning methods and added discussion on the differences with our method (**Appendix D.8**), in response to comments from **Reviewer QvDA, WtCd and pH7N**.
5. We added comparisons to Denoised MDP (**Appendix D.9**), as inquired by **Reviewer WtCd**.
6. We added additional results on two tasks from the Atari domain (**Appendix D.10**) to address a suggestion from **Reviewer pH7N**.
7. We added additional quantitative evaluation of the beneficial impact of observation modeling on reward modeling (**Appendix D.11**).

All updates are highlighted in blue.

Update 11.20: We add additional results on Boxing from the Atari domain.

---

### Meta-Review · Area_Chair_jQ3j · 2023-12-09

**Metareview:**

This paper proposes a new harmonious loss function that balances between explicit and implicit MBRL, and therefore observation and reward modeling pressures that exist in the multi-task setting, purporting to  unify the two approaches in a dynamic way.

The reviewers overall agree the paper is well-written and presented, and find the topic interesting. Their main criticisms center around the strength of the experiments, with suggestions for more comparisons and tasks. No reviewer appeared to feel particularly strongly, and were mostly not responsive during discussions, despite my and the authors’ repeated entreaties to do so. Therefore I briefly looked over the paper myself and closely read over the authors’ responses.

I agree this is an interesting line of work and the paper read well. However I’m unsure how widely applicable this method can be. In particular, I agree with reviewer pH7N that a wider range of environments was needed, and in particular the ones in the original MuZero and Dreamer V2/3 papers. If HarmonyWM is a strict generalization of these other approaches, why not evaluate on the environments these same approaches were tested on: eg Atari, chess, Minecraft, DMLab, b-suite? Why were only robotic control or navigation tasks used? Granted, there is a small experiment in the appendix on 3 atari games, but it’s not clear if HarmonyWM actually provides a benefit (error bars are overlapping for all but Boxing), and why were only these games selected?

The authors state :

“Regarding the environments Reviewer pH7N proposed, it is beyond our computational resources to conduct experiments on GO, and also unfair since MuZero with MCTS is mainly tailored to board games and unable to handle continuous control, while both Dreamer and HarmonyWM are more general architectures.”

This is a somewhat strange argument to make. If the aim is to show that your approach is an improvement over and more general than existing ones, why choose only tasks that the existing implicit MBRL approaches weren’t designed to handle? And why limit yourself only to a subset of tasks that Dreamer used? b-suite for instance is very lightweight to run. This leads me to believe that these results might not generalize outside of robotic control tasks, undercutting the potential applicability of this approach.

I appreciate the authors’ extensive responses and additional experiments added during the revision period, which have already improved the paper. However, given the above, I don’t think it’s ready for publication at ICLR in its current form.

**Justification For Why Not Higher Score:**

Although an interesting paper, with many strengths, in the end it seems like reviewer concerns were not fully addressed. My decision was based on the limited nature of the environments, as compared with those in the original Dreamer and MuZero papers, as well as the fact that there were much stronger papers in my batch.

**Justification For Why Not Lower Score:**

NA

---

### Decision · Program_Chairs · 2024-01-16

Reject